# Variants in the fetal genome near pro-inflammatory cytokine genes on 2q13 associate with gestational duration

Xueping Liu et al.[#]

The duration of pregnancy is influenced by fetal and maternal genetic and non-genetic factors. Here we report a fetal genome-wide association meta-analysis of gestational duration, and early preterm, preterm, and postterm birth in 84,689 infants. One locus on chromosome 2q13 is associated with gestational duration; the association is replicated in 9,291 additional infants (combined $P = 3.96 \times 10^{-14}$). Analysis of 15,588 mother-child pairs shows that the association is driven by fetal rather than maternal genotype. Functional experiments show that the lead SNP, rs7594852, alters the binding of the HIC1 transcriptional repressor. Genes at the locus include several interleukin 1 family members with roles in pro-inflammatory pathways that are central to the process of parturition. Further understanding of the underlying mechanisms will be of great public health importance, since giving birth either before or after the window of term gestation is associated with increased morbidity and mortality.

Pregnancy in eutherian mammals is characterized by tightly regulated physiological processes to ensure normal fetal development and delivery after a narrowly defined period of gestation[1,2]. A conundrum first posed by Sir Peter Medawar[3] more than 60 years ago is how the semi-allogeneic fetus is protected from attack by the mother's immune system. Compared to many other mammals, humans have a highly invasive placentation process with direct contact with the maternal circulation[1,4], and the immunological paradox of pregnancy continues to be an important research topic. It is well-established that successful gestation depends on numerous mechanisms, some of which involve inflammatory pathways[5]. After conception, an inflammatory phase ensures implantation of the blastocyst in the uterine wall[6]. This is followed by a long anti-inflammatory phase in which the maternal adaptive immune response is dampened to allow the development, growth, and maturation of the fetus. Eventually, a second inflammatory phase results in gradual ripening of gestational tissues, followed by parturition[6]. Many other pathways are dynamically regulated over the course of a pregnancy and are required for the successful completion of pregnancy and timely parturition[7].

Correct timing of parturition is critical for the health of the newborn. Preterm birth, defined as birth before 37 completed weeks of gestation, is not only a major cause of perinatal mortality and morbidity[8], but is also associated with long-term adverse health outcomes including neurodevelopmental delay[9], cerebral palsy[10], diabetes[11], increased blood pressure[12], and various psychiatric disorders[13]. Postterm birth, defined as birth after a gestation of 42 completed weeks (hereafter weeks) or more is associated with increased risks of fetal and neonatal mortality and morbidity plus increased maternal morbidity[14]. Each of these outcomes affects approximately 5% to 10% of all births in high income countries[15,16] and preterm birth rates are considerably higher in some low- and middle-income countries[17].

Although timing of parturition is influenced by many non-genetic risk factors, including parity, maternal stress, smoking, urogenital infection, educational attainment, and socioeconomic status, there is compelling evidence for a substantial genetic impact[18,19]. For example, twin and family studies have estimated that the heritability of gestational duration ranges from 25% to 40%[20]. Several studies have shown that the duration of pregnancy has both heritable maternal and fetal components[21,22]. Estimates from a Swedish family study analyzing 244,000 births indicated that fetal genetic factors explained about 10% of the variation in gestational duration, whereas maternal factors accounted for about 20%[22].

Little is known about specific fetal and maternal genetic contributions to gestational duration. Most genome-wide association studies (GWAS) of birth timing have been limited in sample size and have not identified robustly associated genetic loci[23–27]. Recently, however, a GWAS based on samples from 43,568 women of European ancestries identified maternal genetic variants at six loci associated with gestational duration at $P < 5 \times 10^{-8}$ with replication in three independent data sets[28]. Three of these loci were also associated with preterm birth as a dichotomous outcome.

In the current study, our goal is to identify fetal genetic variants associated with timing of parturition. We conduct a GWAS meta-analysis of gestational duration as a quantitative trait and of the clinically relevant dichotomous outcomes early preterm (<34 weeks), preterm (<37 weeks), and postterm (≥42 weeks) birth, in 84,689 infants from cohorts included in the Early Growth Genetics (EGG) Consortium, the Initiative for Integrative Psychiatric Research (iPSYCH) study, and the Genomic and Proteomic Network for Preterm Birth Research (GPN), with replication analyses in 9291 infants from additional cohorts. Since a child inherits half of its genetic material from its mother, their genotypes at a given locus are highly correlated, and it may not be clear whether a genetic association reflects the effect of the child's own genotype on the timing of their delivery, or an effect of their mother's genotype on the timing of parturition. For 15,588 of the infants, maternal genetic data are also available, allowing us to address whether identified genetic effects are of fetal or maternal origin.

## Results

**Discovery stage.** Characteristics of the 20 studies included in the discovery stage are presented in Supplementary Data 1. The discovery data set included information on 84,689 infants, 4775 of whom were born preterm (<37 weeks), with 1139 of these considered early preterm infants (<34 weeks). A further 60,148 infants were born ≥39 weeks and <42 weeks of gestation and were used as term controls. Finally, 7888 infants were born postterm (≥42 weeks). Our study design is illustrated in Supplementary Fig. 1. After imputation using reference data from the Haplotype Reference Consortium release 1.1 (ref. [29]) or the integrated phase III release of the 1000 Genomes Project[30], each contributing study performed GWAS analyses for at least one of the four study traits, assuming an additive genetic model (see Methods for details). Final meta-analysis results were obtained for >7.5 million SNPs for each of gestational duration, early preterm birth, preterm birth, and postterm birth, with genomic inflation factors <1.05. Quantile–quantile and Manhattan plots for the four phenotypes are shown in Supplementary Fig. 2. In the discovery GWAS meta-analysis, one locus (on chromosome 2q13) was associated with gestational duration and postterm birth at genome-wide significance ($P < 5 \times 10^{-8}$) (Supplementary Fig. 2A, B), and two loci (on chromosomes 1p33 and 3q28) were significantly associated with early preterm birth (Supplementary Fig. 2C). No locus reached genome-wide significance for preterm birth (Supplementary Fig. 2D). We selected one lead SNP for each of the three loci reaching genome-wide significance for analysis in the replication stage.

**A locus harboring pro-inflammatory cytokine genes.** At the 2q13 locus, rs7594852 was the SNP most significantly associated with gestational duration ($P = 1.88 \times 10^{-12}$; Fig. 1a) and was selected as the lead SNP for replication stage analysis. For postterm birth, we also selected rs7594852 ($P = 4.64 \times 10^{-8}$, Fig. 1b) as the lead SNP for the replication analyses from a set of highly correlated SNPs ($r^2 > 0.98$) with very similar P values. The association of rs7594852 with gestational duration was replicated, with a P value of $3.69 \times 10^{-3}$ in the replication sample and an overall P value of $3.96 \times 10^{-14}$ in the combined discovery and replication analysis (Table 1). In the combined analysis, each additional fetal rs7594852-C allele was associated with an additional 0.37 days (95% confidence interval (CI) = 0.22−0.51) of gestational duration. For postterm birth the statistical power to replicate the association was modest at 40% (Supplementary Table 1) and the SNP did not reach nominal significance in the replication stage analysis, although the direction of the effect was consistent with the discovery stage (Table 1). rs7594852 is intronic in CKAP2L and is located in a linkage disequilibrium (LD) block that encompasses IL1A, IL1B, and several other genes encoding proteins in the interleukin 1 cytokine family (Fig. 1a, b). In an additional analysis conditioning on rs7594852, we found no evidence for multiple independent signals at the locus (Supplementary Fig. 3). Figure 2 shows a forest plot of association results for rs7594852 across all studies. The estimated variance in gestational duration explained by rs7594852 was 0.066%.

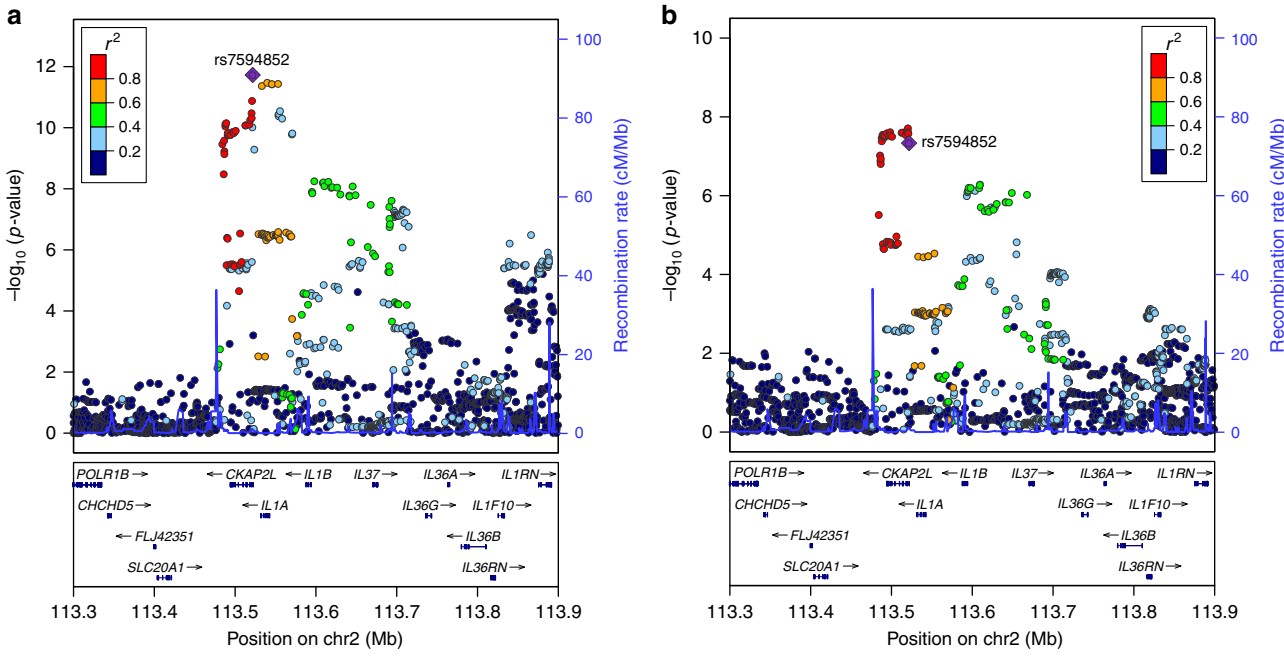

**Fig. 1** Discovery stage results at the 2q13 locus. Regional association plots for **a** gestational duration and **b** postterm birth. SNP position is shown on the *x*-axis and association ($-\log10$ $P$ value) with gestational duration and postterm birth, respectively, on the left *y*-axis. The lead SNP, rs7594852, from the gestational duration analysis is represented by a purple diamond, and the other SNPs are colored to reflect their LD with the lead SNP (based on pairwise $r^2$ values from the Danish National Birth Cohort). In the postterm birth analysis, rs7607470 had a slightly lower $P$ value, but this SNP is highly correlated with rs7594852 ($r^2 > 0.99$), and the latter SNP was selected for the replication stage analyses of both gestational duration and postterm birth. Estimated recombination rates are from HapMap (right *y*-axis)

| | | rs7594852 effect allele (C) frequency | | Number | | | | | |
|---|---|---|---|---|---|---|---|---|---|
| **Table 1 Discovery, replication, and combined results for the lead SNP rs7594852 at the 2q13 locus** | | | | | | | | | |
| **Phenotype** | **Sample sets** | **Cases** | **Controls** | **Cases** | **Controls** | **Beta/OR[a] (95% CI)** | **P** | **I[2] (95% CI)** | **P_{het}** |
| Gestational duration | Combined discovery | | 0.53 | | 84,689 | 0.034 (0.024–0.043) | $1.88 \times 10^{-12}$ | 0 (0.0–13.4) | 0.97 |
| | MoBa_HARVEST | | 0.54 | | 7072 | 0.049 (0.016–0.082) | $3.78 \times 10^{-3}$ | | |
| | BiB | | 0.51 | | 1354 | 0.058 (−0.028–0.144) | 0.18 | | |
| | FIN | | 0.58 | | 865 | 0.004 (−0.056–0.065) | 0.89 | | |
| | Combined replication | | | | 9291 | 0.041 (0.013–0.068) | $3.69 \times 10^{-3}$ | 0 (0.0–97.4) | 0.41 |
| | All combined | | | | 93,980 | 0.034 (0.025–0.043) | $3.96 \times 10^{-14}$ | 0 (0.0–99.6) | 0.64 |
| Postterm birth | Combined discovery | 0.55 | 0.53 | 7888 | 52,807 | 1.1 (1.06–1.14) | $4.64 \times 10^{-8}$ | 7.3 (0.0–88.1) | 0.37 |
| | MoBa_HARVEST | 0.55 | 0.54 | 670 | 5626 | 1.05 (0.89–1.24) | 0.39 | | |
| | All combined | | | 8558 | 58,433 | 1.1 (1.06–1.14) | $4.34 \times 10^{-8}$ | 0 (0.0–99.7) | 0.58 |

[a]For the quantitative trait of quantile transformed gestational duration, the column reports beta estimates. For the dichotomous trait postterm birth, odds ratio (OR) estimates are given; CI, confidence interval; $I^2$, heterogeneity estimate (proportion of variance that is due to between study differences); $P_{het}$, $P$ value from the Cochran Q test of between study heterogeneity. Individual study association $P$ values are two-sided and obtained by linear regression (for quantile transformed gestational duration) or logistic regression (for postterm birth). Combined $P$ values are also two-sided and obtained from fixed-effects inverse-variance-weighted meta-analysis

No association was seen at the 2q13 locus in case–control analyses of early preterm birth (odds ratio (OR) = 1.02, 95% CI = 0.93–1.12, $P = 0.68$ for rs7594852-C) or preterm birth (OR = 0.96, 95% CI = 0.92–1.00, $P = 0.07$ for rs7594852-C; see also Supplementary Fig. 4). This may suggest that other mechanisms could be playing a role in causing early parturition before the mechanisms mediating the effect of the locus get the opportunity to influence the phenotype. To further investigate this question, we binned the 51,357 births from the largest contributing study (iPSYCH) in five groups by gestational duration. We then estimated the frequency of the rs7594852-C allele in each group and in the whole sample. In the overall meta-analysis, each additional fetal rs7594852-C allele was associated with increased gestational duration (Table 1). The frequency of the rs7594852-C allele in the group with the shortest gestational duration was only slightly lower than the frequency in the whole sample (Supplementary Fig. 5). The lowest allele frequency (0.518) was seen in the second group, representing a mean gestational duration of 276.5 days. The allele frequency then gradually increased in the next groups with the highest frequency (0.555) observed for the group representing the longest gestational duration (mean of 298.3 days) (Supplementary Fig. 5). This pattern in allele frequencies deviates from what is expected under the hypothesis that the strength of the association is independent of gestational duration ($P = 0.0013$, semi-parametric bootstrap, see Supplementary Methods for details).

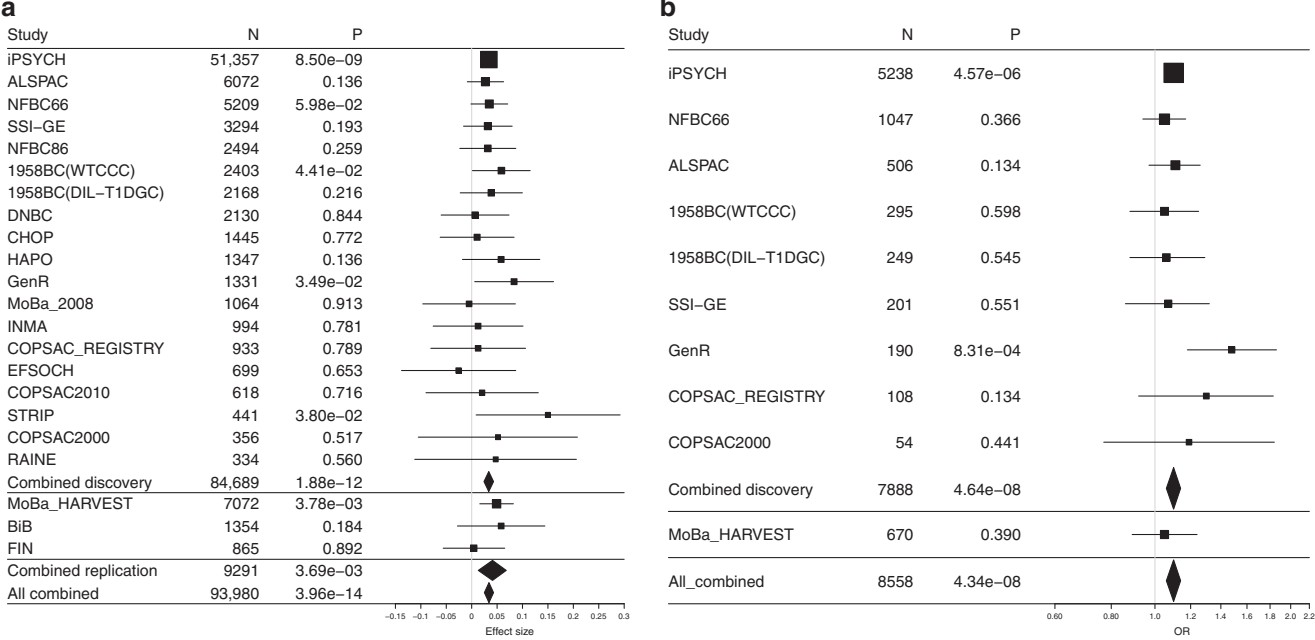

**Fig. 2** Forest plots showing association results for rs7594852. **a** Gestational duration effect estimates with 95% CIs, and **b** postterm birth ORs with 95% CIs. Source data are provided as a Source Data file

To explore possible functional mechanisms underlying the association signal at the 2q13 locus, we annotated the 283 variants within 1 Mb of the lead SNP that were associated with gestational duration with $P < 1 \times 10^{-4}$ (Supplementary Data 2). Among these, six variants were categorized as exonic (two synonymous (in *CKAP2L* and *IL1RN*) and four missense variants (all in *CKAP2L*, see Supplementary Data 3). Additionally, 190 of the 283 variants have been reported by the GTEx[31] and GEUVA-DIS[32] consortia as cis-eQTLs ($P < 1 \times 10^{-4}$) for several nearby genes, including *IL1A*, *IL1B*, *IL36B*, and *IL1RN*, in different tissues. Of note, the lead variant rs7594852-C allele is associated with decreased expression of *IL1A* in skin ($P = 7.36 \times 10^{-6}$) and decreased expression of *IL1B* in lymphoblastoid cell lines ($P = 4.41 \times 10^{-6}$). Among the 283 top variants at the 2q13 locus, 104 are located in likely enhancer regions of (mainly) cytokine genes (see Methods). GWAS Catalog[33] annotation revealed that the variant rs10167914 ($P = 2.66 \times 10^{-7}$ for gestational duration, $r^2 = 0.64$ with rs7594852) has been reported to be associated with endometriosis[34] ($P < 5 \times 10^{-8}$), with the risk allele for endometriosis corresponding to increased gestational duration in our data (Supplementary Data 2).

Exome sequencing data were available for 18,382 subjects from the iPSYCH study. To investigate whether any single exonic variant could explain the observed association at the 2q13 locus, we tested all exonic variants with allele count larger than two in a 1 MB region around the lead SNP rs7594852 for association with gestational duration (see Methods). Among 272 exonic variants tested, seven were associated with gestational duration at $P < 0.01$, with the lowest $P = 0.00018$. These variants were, however, either not in very high LD with rs7594852 ($r^2 < 0.1$) or did not remain associated after conditioning on rs7594852 genotype (Supplementary Data 4). For all genes in the same 1 MB region, we further carried out gene-based tests for aggregated effects of rare coding variants (optimal sequence kernel association test, SKAT-O; see Methods) now also including variants only observed with allele counts of one or two. Among the 18 genes tested, we were not able to detect any significant gene-based rare-variant association to gestational duration (Supplementary Data 4).

Thus, we found no exonic variants likely to explain the observed association.

**Early preterm birth associations**. Our early preterm birth meta-analysis revealed two genome-wide significant loci in the discovery stage. At the first locus on chromosome 3q28, rs112912841 yielded the lowest $P$ value (OR = 1.64, 95% CI = 1.38–1.94, $P = 9.85 \times 10^{-9}$). This SNP is intronic in *LPP* (Supplementary Fig. 6A) and was nominally significantly associated with preterm birth (OR = 1.12, 95% CI = 1.02–1.23, $P = 0.02$) but not with gestational duration ($P = 0.24$). The most significant SNP at the second locus on chromosome 1p33, rs1877720 (OR = 1.64, 95% CI = 1.37–1.96, $P = 4.33 \times 10^{-8}$), is intronic in *SPATA6* (Supplementary Fig. 6B) and was nominally significantly associated with preterm birth (OR = 1.22, 95% CI = 1.11–1.34, $P = 4.31 \times 10^{-5}$) and gestational duration ($P = 2.95 \times 10^{-5}$). There was no evidence for multiple independent signals at these loci when the lead SNP was included as a covariate in conditional regression analyses (Supplementary Fig. 7). We conducted replication analyses for the two lead SNPs (rs112912841 and rs1877720) in an independent Finnish case–control study (107 infants born early preterm and 865 born at term), but statistical power was low (Supplementary Table 1) and we could not confirm the associations (Supplementary Fig. 8 and Supplementary Table 2). We also conducted family-based association tests of the lead SNPs at these two loci in a set of 276 early preterm birth trios of European ancestries from Iowa, but could not find support for the signals in that data set either (Supplementary Table 3).

**Fetal or maternal genetic effects?** Gestational duration is a complex outcome influenced by both the maternal and fetal genomes. To disentangle fetal and maternal genetic effects, we first compared our fetal association results with those from a recent maternal GWAS of gestational duration[28]. In our fetal analysis, each C allele of the lead variant, rs7594852, was associated with an additional 0.37 days (95% CI = 0.22−0.51) of gestational duration (estimate based on 51,357 infants from the iPSYCH study, see Methods). In the published maternal GWAS

($n = 43,568$), the direction of effect was the same: each maternal rs7594852-C allele was associated with an additional 0.22 days (95% CI $= -0.01-0.45$) of gestational duration, though the confidence intervals were wide and included the null value. The fact that the maternal effect estimate was approximately half the size of the fetal effect estimate (0.22 vs. 0.37 days) is as expected if the association is of fetal origin. Also, we used a recently developed weighted linear model (WLM) approach[35] to obtain an estimate of the fetal effect adjusted for the maternal genotype and vice versa (see Methods). The WLM-adjusted fetal effect was 0.34 days (95% CI $= 0.09-0.59$, $P = 6.60 \times 10^{-3}$, two-sided Z-test) close to the unadjusted estimate of 0.37 days, whereas the WLM-adjusted maternal effect was 0.05 days (95% CI $= -0.27$ to 0.37, $P = 0.77$). Next, we performed joint maternal–fetal genetic association analyses of the lead variant, rs7594852, at the 2q13 locus in 15,588 mother–child pairs that met the inclusion criteria from the discovery stage (see Methods for details). Here we found that conditioning on maternal genotype did not attenuate the fetal genetic association while the maternal genetic association conditional on fetal genotype was non-significant (Table 2). Taken together, these results indicate that the association signal for gestational duration at the 2q13 locus represents a fetal genetic effect. Conversely, we examined the lead variants at four of the six loci, which were reported to be significant in the maternal GWAS and were available in our meta-analysis (the remaining two were not autosomal)[28]. We found evidence of association in the fetal genome at the *EBF1* ($P = 1.18 \times 10^{-6}$), *EEFSEC* ($P = 0.05$), *WNT4* ($P = 5.37 \times 10^{-5}$), and *ADCY5* ($P = 0.005$) loci. For all four loci, the direction of effects in the fetal GWAS was consistent with the published maternal GWAS results, but fetal effect size estimates were smaller (Supplementary Data 5). Conditional analyses in the 15,588 mother–child pairs were also consistent with effects at these four loci being attributable to the maternal genome (Supplementary Data 5).

**Heritability and genetic correlation with other traits.** Based on the gestational duration summary statistics for all common autosomal SNPs (minor allele frequency, MAF >1%), the estimated proportion of variance explained (SNP heritability) was 7.6% (SE $= 0.8$%). This estimate was based on the quantile transformed phenotype, and when using results for gestational duration in days (based on 51,357 infants from the iPSYCH study) the variance explained was 4.5% (SE $= 1.1$%). For comparison, we analyzed summary statistics from a recent maternal GWAS of gestational duration in days (based on 43,568 mothers)[28] using the same SNP set and found that the proportion of variance explained was 7.9% (SE $= 1.5$%). However, the above estimates are all influenced by fetal as well as maternal genetic loci. To obtain estimates of fetal effect adjusted for the maternal genotype and vice versa for each SNP, we combined the unadjusted fetal effects with unadjusted maternal effects (based on gestational duration in days) using the WLM approach[35]. The proportion of variance explained by WLM-adjusted fetal effects was 1.3% (SE $= 1.0$%) and that of WLM-adjusted maternal effects was 4.9% (SE $= 1.3$%).

As expected there was a strong positive correlation between unadjusted fetal and maternal effects for gestational duration in days ($r_g = 0.77$, SE $= 0.17$, $P = 4.29 \times 10^{-6}$, two-sided Z-test). When performing genetic correlation analyses between our meta-analysis results for (quantile transformed) gestational duration and 690 traits and diseases in LDHub[36], we found that eight were significant after correction for multiple testing (Supplementary Data 6). These included positive genetic correlations with own birth weight ($r_g = 0.21$, SE $= 0.04$, $P = 4.14 \times 10^{-6}$) and birth weight of first child ($r_g = 0.28$, SE $= 0.05$, $P = 1.23 \times 10^{-8}$).

However, when instead using WLM-adjusted fetal effects, no correlations remained significant after Bonferroni correction. For WLM-adjusted maternal effects, there was a positive genetic correlation with birth weight of first child ($r_g = 0.50$, SE $= 0.095$, $P = 1.16 \times 10^{-7}$; Supplementary Data 6), in line with recent findings[35].

**Functional analyses.** Exome sequencing data and variant annotation of the 2q13 locus did not identify any exonic variants (alone or combined by gene) likely to explain the association with gestational duration, suggesting that the underlying mechanism might instead involve altered gene regulatory mechanisms. To investigate potential mechanisms, we conducted a series of functional experiments and analyses. We first prioritized all SNPs in the LD block containing the association signal based on their overlap with functional genomics data sets from cell types relevant to gestation (see Methods). The highest ranked variant was the discovery lead variant rs7594852, which overlaps with 17 different data sets, with no other variant intersecting more than three (Supplementary Data 7). We then used the Cis-BP database[37] to identify transcription factors that might bind differentially to this variant. These analyses revealed that the rs7594852-C allele might alter the binding of the hypermethylated in cancer 1 (HIC1) protein (Fig. 3a). The HIC1 protein is a $C_2H_2$ zinc-finger transcriptional repressor with a consensus DNA binding sequence containing a core GGCA motif[38]. Protein-binding microarray enrichment scores (E-scores) indicate strong binding of HIC1 to the cytosine C allele (E-score $= 0.48$), and only moderate binding to the alternative thymidine (T) allele (E-score $= 0.32$). We confirmed the presence of multiple histone marks overlapping rs7494852 in various fetal cell and tissue types (chorion, amniocytes, trophoblasts)[39], indicating that the chromatin in this locus is likely accessible and active in these fetal cells (Fig. 3b). In particular, the fetal side of the placenta displays a strong H3K4me3 modification signal, a histone mark often found in active regulatory regions. Using an electrophoretic mobility shift assay, we detected enhanced binding of HIC1 to the rs7594852- C allele (Fig. 3c), as predicted. Densitometric analysis for HIC1 band intensity demonstrated a statistically significant difference in average intensity between the rs7594852-C (mean $= 204.0$, SD $= 60.0$) and rs7594852-T (mean $= 76.6$, SD $= 42.3$) alleles respectively ($P = 0.013$, Student's t-test, $n = 4$ per group). Additional experiments are needed to investigate which of the genes at the 2q13 locus have altered transcriptional levels in relevant cell types due to the enhanced binding of HIC1 to the rs7594852-C allele.

Next, we examined associations of the lead SNP, rs7594852 with gene expression using RNA sequencing data from 102 human placentas[40]. Here we found that among 118 genes/ transcripts with transcription start sites within 500 kb of rs7594852, there were five nominally significant ($P < 0.05$) cis-eQTL associations (Supplementary Table 4). Three of these genes (*IL1A*, *IL36G*, *IL36RN*) encode proteins in the interleukin 1 cytokine family. The rs7594852-C allele, which in our data was associated with increased gestational duration, corresponded to decreased expression of the cytokine-encoding genes *IL1A* and *IL36G*. Conversely, the rs7594852-C allele corresponded to increased expression of *IL36RN*, which encodes an antagonist to the interleukin-36-receptor.

Finally, to evaluate a possible general effect of the 2q13 locus on inflammatory markers we tested for associations between the lead SNP rs7594852 and levels of inflammatory markers in peripheral blood from newborns, using data from the iPSYCH study. None of the cytokines encoded by genes at the 2q13 locus had been assayed, but measurements of the biomarkers BDNF, CRP, EPO, IgA, IL8, IL-18, MCP1, S100B, TARC, and VEGFA

**Table 2 Associations in 15,588 mother–child pairs between rs7594852 genotype and gestational duration**

| Study | N | Fetal effect (unadjusted for maternal genotype) | | Fetal effect (adjusted for maternal genotype) | | Maternal effect (unadjusted for fetal genotype) | | Maternal effect (adjusted for fetal genotype) | |
|---|---|---|---|---|---|---|---|---|---|
| | | Beta (95% CI) | P | Beta (95% CI) | P | Beta (95% CI) | P | Beta (95% CI) | P |
| MoBa_HARVEST | 6362 | 0.041 (0.001, 0.076) | 0.020 | 0.051 (0.011, 0.092) | 0.013 | 0.0065 (−0.029, 0.041) | 0.72 | −0.020 (−0.060, 0.021) | 0.34 |
| ALSPAC | 4305 | 0.028 (−0.013, 0.07) | 0.18 | 0.021 (−0.027, 0.07) | 0.39 | 0.025 (−0.017, 0.067) | 0.24 | 0.014 (−0.034, 0.063) | 0.57 |
| DNBC | 1396 | −0.016 (−0.09, 0.059) | 0.68 | −0.009 (−0.094, 0.077) | 0.84 | −0.018 (−0.090, 0.054) | 0.63 | −0.014 (−0.097, 0.069) | 0.75 |
| BiB | 1182 | 0.09 (−0.002, 0.18) | 0.055 | 0.090 (−0.015, 0.19) | 0.09 | 0.045 (−0.049, 0.14) | 0.35 | 0.00065 (−0.11, 0.11) | 0.99 |
| MoBa_2008 | 854 | −0.040 (−0.139, 0.058) | 0.42 | −0.026 (−0.138, 0.085) | 0.64 | −0.041 (−0.137, 0.056) | 0.41 | −0.029 (−0.138, 0.080) | 0.61 |
| FIN | 833 | −0.012 (−0.073, 0.050) | 0.57 | 0.021 (−0.053, 0.094) | 0.58 | −0.005 (−0.057, 0.066) | 0.88 | −0.029 (−0.10, 0.045) | 0.44 |
| EFSOCH | 656 | −0.037 (−0.147, 0.073) | 0.51 | 0.028 (−0.098, 0.154) | 0.67 | −0.114 (−0.221, −0.007) | 0.037 | −0.128 (−0.251, −0.005) | 0.043 |
| All combined | 15588 | 0.022 (0.001, 0.044) | 0.042 | 0.032 (0.007, 0.057) | 0.012 | 0.004 (−0.018, 0.025) | 0.73 | −0.015 (−0.040, 0.010) | 0.24 |

N indicates number of complete mother–child pairs (i.e. where genotype data were available for both mother and child); Beta is estimated under an additive model with rs7594852-C as the effect allele; CI, confidence interval. Individual study P values are two-sided and obtained by linear regression of quantile transformed gestational duration. Combined P values are also two-sided and obtained from fixed-effects inverse-variance-weighted meta-analysis

were available for 8138 iPSYCH samples. However, we found no significant associations between rs7594852 genotype and levels of these biomarkers (Supplementary Table 5). We also used the GWAS Catalog[33] to identify 39 SNPs known from previous studies to be associated with cytokine levels and examined associations for these SNPs with gestational duration in our discovery stage meta-analysis. These SNPs did not show more evidence of association with gestational duration than expected by chance (Supplementary Fig. 9).

### Discussion

In this genome-wide meta-analysis including a total of 93,980 infants in the discovery and replication stages, we identified a fetal locus on chromosome 2q13 that was robustly associated with gestational duration. The lead SNP at the locus, rs7594852, was also associated with postterm birth as a dichotomous outcome in the discovery stage, but not with either preterm birth or early preterm birth. An analysis of allele frequency in different strata of the gestational duration distribution confirmed that genetic variation at the locus was most strongly associated with timing of parturition in later stages of pregnancy.

Gestational duration is a complex phenotype influenced by both fetal and maternal genetic contributions, as well as environmental factors. Since mothers and children share half of their genetic material, we investigated the possibility that the 2q13 association signal could represent a maternal, rather than a fetal, effect. This was not the case: our analysis of more than 15,000 mother–child pairs showed that the association had a fetal origin independent of the maternal genotype. Furthermore, the lead SNP, rs7594852, was not significantly associated with gestational duration in a maternal GWAS including more than 40,000 women[28] and WLM-adjusted estimates also supported the fetal origin of the association.

Looking across the genome, we found that common autosomal fetal genetic variants explained 7.6% of the variance in (quantile transformed) gestational duration. When instead analyzing gestational duration in days (untransformed, based on 51,357 infants from the iPSYCH study), the fraction of variance explained by common fetal variants was 4.5%. However, to fully address the question of variance explained by fetal genetic

variation, the maternal genetic contribution needs to be accounted for. Combining fetal results for gestational duration in days with corresponding maternal results from an independent sample[28] using the WLM approach, the fraction of variance explained was 1.3% for WLM-adjusted fetal effects and 4.9% for WLM-adjusted maternal effects. The larger influence of maternal compared to fetal genetic variation on gestational duration is consistent with findings from large family studies in populations of Scandinavian origin[21,22], but our estimates of variance explained are lower, both before and after WLM adjustment. Such missing heritability has been observed for many traits and diseases, and is often attributed to rare causal variants in low LD with common SNPs as well as possible overestimation of heritability in family studies due to shared environmental effects or non-additive genetic variation[41]. Furthermore, we note that giving more weight to observations in the lower tail of the distribution (by going from the quantile transformed phenotype to the untransformed phenotype) resulted in lower heritability estimates. This was also observed in a large family study, which therefore excluded births before week 35 when estimating heritability[21]. A detailed dissection of fetal and maternal contributions to the heritability of gestational duration lies beyond the scope of the current study, but is an important topic for future research.

While the lead SNP at the new 2q13 locus is intronic in CKAP2L, which encodes a mitotic spindle protein, the locus also harbors a number of genes encoding proteins in the interleukin 1 family of pro-inflammatory cytokines. It is well-established that IL-1 signaling plays a central role in the process leading to parturition in healthy term pregnancies[42]. However, infections or trauma can also induce increased secretion of IL-1 and other pro-inflammatory cytokines, provoking preterm birth[42]. In our data, genetic variation at the 2q13 locus was most strongly associated with gestational duration in later stages of pregnancy. We hypothesize that this locus is involved in genetic regulation of a pro-inflammatory cytokine signaling mechanism by which the mature fetus communicates to the mother that it is ready to be born. In a first step towards understanding the molecular mechanisms underlying the genetic association, we found that the rs7594852-C allele creates a strong binding site for the transcriptional repressor HIC1. It is conceivable that the variant

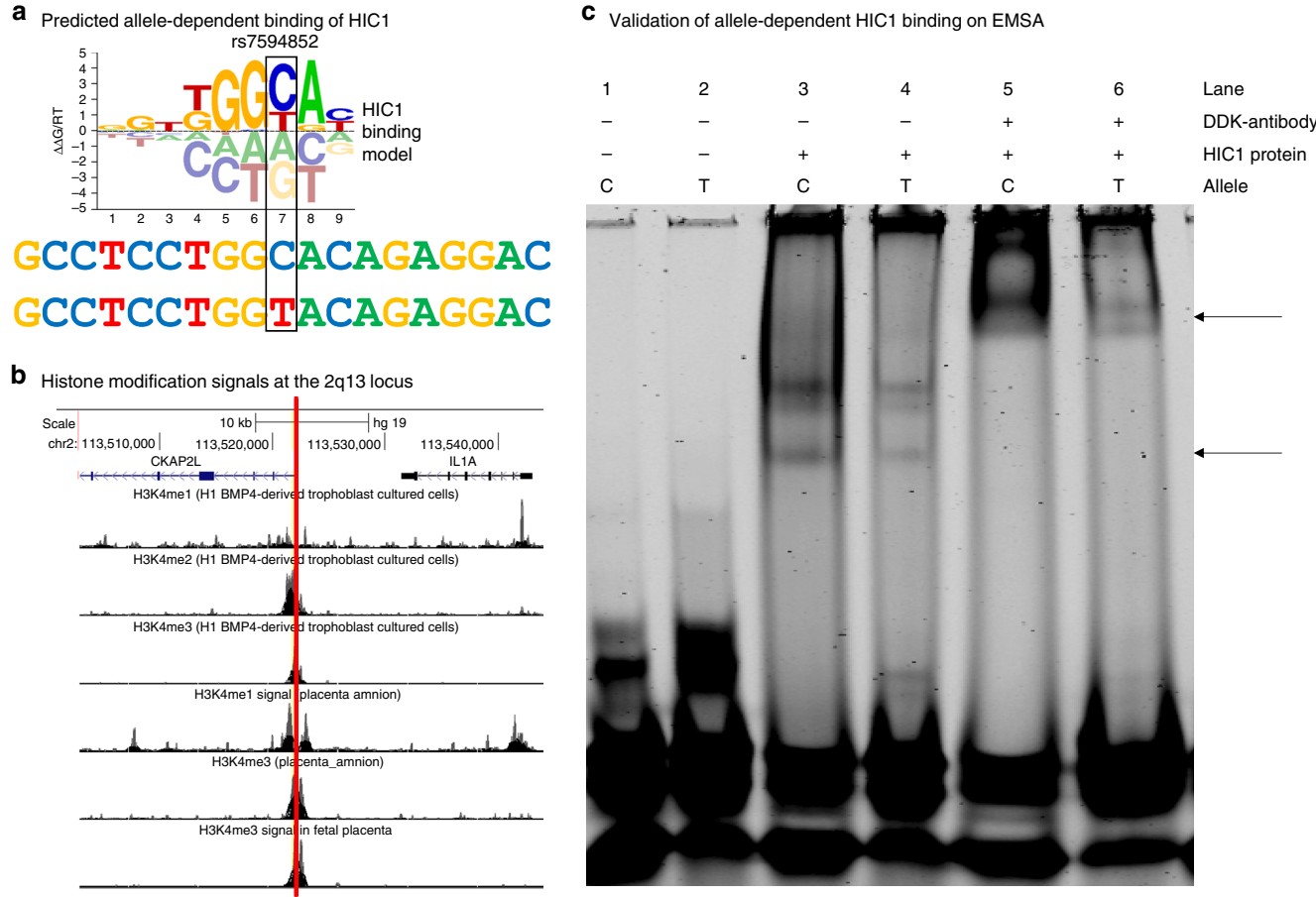

**Fig. 3** HIC1 binding at the 2q13 locus. **a** The rs7594852-C allele creates a stronger binding site for the hypermethylated in cancer 1 (HIC1) protein. The sequence logo of the HIC1-binding motif shows the DNA binding preferences of HIC1. Tall nucleotides above the dashed line indicate DNA bases that are preferred by HIC1, whereas bases below the dashed line are disfavored. The *y*-axis indicates the relative free energies of binding for each nucleotide at each position. The height of each nucleotide can be interpreted as the free energy difference from the average ($\Delta\Delta G$) in units of gas constant (*R*) and temperature (*T*). The DNA sequence flanking the rs7594852-C allele is shown directly below, with the alternative rs7594852-T allele shown at the bottom. The rs7594852-T allele changes the HIC1 binding site sequence from C (most favored) to T (less favored). **b** UC Santa Cruz Genome Browser screenshot depicting the rs7594852 locus. The purple (*CKAP2L*) and black (*IL1A*) graphics at the top indicate the locations of exons (columns), untranslated regions (rectangles), and introns (horizontal lines), with arrows indicating the direction of transcription. The red vertical line indicates the position of rs7594852, which overlaps strong signals obtained from chromatin immunoprecipitation sequencing (ChIP-seq) experiments (indicating histone modification by mono-, di-, or trimethylation of histone H3 on lysine 4; H3K4) in trophoblast cultured cells, placental amnion, or fetal placenta. **c** Experimental validation of allele-dependent binding of human purified recombinant HIC1 protein with a c-Myc/DDK tag to rs7594852 and flanking sequence via electrophoretic mobility shift assay (EMSA). Arrows indicate allele-dependent binding of HIC1 (bottom arrow) and a supershift of the protein–DNA complex induced by the binding of the anti-DDK antibody to the complex (top arrow). The presence of multiple bands in lanes 3 and 4 is likely due to the presence of multiple HIC1 isoforms. Source data are provided as a Source Data file

thereby delays signaling from the fetus to the mother, thus prolonging gestation, possibly until other (redundant) mechanisms stimulating parturition take effect. We had limited statistical power to directly evaluate effects of the variant allele on expression levels of genes at the locus, but the results of eQTL analyses in 102 placental samples collected after birth were compatible with the hypothesis that the rs7594852-C allele may lead to prolonged gestational duration through decreased gene expression of one or more members of the interleukin 1 cytokine family genes. eQTL results from the GTEx and GEUVADIS databases for skin tissue and lymphoblastoid cell lines, respectively, support this suggestion. However, the possible link between enhanced binding of HIC1 to the rs7594852-C allele and altered expression of one or more genes at the 2q13 locus in relevant cell types still needs to be investigated and our functional analyses do not rule out other potential mechanisms. To fully illuminate the biological mechanisms underlying our genetic association results larger

sample sizes and further functional follow-up experiments are needed, including fine-mapping of the locus and characterization of pro-inflammatory cytokine signaling shortly before parturition, e.g., through non-invasive techniques allowing quantification of cell-free fetal RNA[43,44] and measurement of cytokine levels.

The fact that the 2q13 locus was not associated with preterm birth or early preterm birth in our case–control analyses underlines that genetic triggers of parturition probably change as gestation advances. Genetic analysis of preterm birth is further complicated by the possible influence of a wide range of environmental factors. Population-based heritability analyses of gestational duration demonstrate that inclusion of early preterm births results in decreased estimates of the fetal genetic effect[21] and previous GWAS efforts have not identified robust fetal genetic associations with preterm birth[23–25]. To refine outcome definitions, we applied a range of exclusion criteria, but although our analyses were based on almost 5000 infants born preterm,

with 1139 of these considered early preterm births, no loci were robustly associated with preterm birth. Statistical power calculations suggested that our preterm birth analyses were well-powered to detect associations with common (MAF > 0.1) fetal genetic variants with odds ratios >1.25 (Supplementary Fig. 10). Fetal genetic contributions to preterm birth may therefore involve smaller effect sizes or less frequent variants. Studies with larger sample sizes would be needed to address this question.

Our study had limitations, including the restriction to infants of European descent. Further studies are therefore warranted to delineate fetal and maternal genetic contributions to gestational duration in populations of non-European ancestries. A second limitation of our study was the differences in gestational duration ascertainment from study to study. In many of the older cohorts that recruited women who were pregnant prior to routine use of ultrasound scan dating, gestational duration estimates were based largely on maternal-reported last menstrual period at the time of pregnancy, whereas estimates for infants born more recently were predominantly based on first-trimester ultrasound screening (Supplementary Data 1). Also, while the overall fraction of preterm births in the discovery stage was 4775/84,689 = 5.6%, the contributing studies included case/control studies of preterm birth (with ~40–50% cases), birth cohorts (with more population representative fractions of preterm births), and cohorts where preterm births were not included (Supplementary Data 1). Furthermore, the degree to which children were excluded based on maternal conditions or pregnancy complications differed among cohorts. However, while these sources of heterogeneity may have caused some underestimation of effect sizes at genuinely associated loci, it should not have resulted in increased false-positive rates. Our extensive exclusion criteria aimed to focus on "natural" gestational duration rather than specific causes such as preterm birth due to pregnancy complications, assisted delivery, or congenital anomalies. Although such exclusions can result in selection bias[45], the overall consistency between studies (Table 1, Fig. 2), despite their varying ability to completely apply all of our pre-specified exclusion criteria, provides some reassurance that any such bias may not be large. One might also speculate as to whether spurious signals could arise from the case groups of various diseases that were included in the discovery stage analyses. However, we consider this unlikely since the association analyses were stratified by disease group and since we did not observe heterogeneity of effect estimates between studies of various design (including population-based cohorts) for the 2q13 lead variant (Table 1, Fig. 2).

In conclusion, parturition is a complex physiological process involving multiple redundant mechanisms influenced by maternal and fetal factors[2]. An enhanced understanding of these mechanisms is of great public health importance, since giving birth either before or after the window of term gestation is associated with increased morbidity and mortality[8,14]. Our study identified the first robustly associated fetal genetic locus for gestational duration. The effect was observed in pregnancies that went to term or beyond and our results raise the hypothesis that variants at the associated locus influence the regulation of pro-inflammatory cytokines in the IL-1 family. Our findings provide a foundation for further functional studies that are required to refine our understanding of the biology of the timing of parturition.

## Methods

**Discovery stage cohorts**. Analyses were performed among participants of studies in the Early Growth Genetics (EGG) Consortium, the Initiative for Integrative Psychiatric Research (iPSYCH) study, and the Genomic and Proteomic Network for Preterm Birth Research (GPN). The iPSYCH sample (n = 51,357) included patient groups of six mental disorders: autism, ADHD, schizophrenia, bipolar disorder, depression, and anorexia[46]. Participating studies from the EGG Consortium included the Avon Longitudinal Study of Parents and Children (ALSPAC, n = 6072) study, the Children's Hospital of Philadelphia (CHOP, n = 1445) cohort, three sub-samples from the Copenhagen Prospective Studies on Asthma in Childhood (COPSAC_REGISTRY, n = 933; COPSAC2000, n = 356; COPSAC2010, n = 618), a sub-sample from the Danish National Birth Cohort (DNBC, n = 2,130), the Exeter Family Study of Children's Health (EFSOCH, n = 699) study, the GENERATION R (GenR, n = 1331) study, the Hyperglycemia and Adverse Pregnancy Outcome (HAPO, n = 1347) study, the Infancia y Medio Ambiente (INMA, n = 994) study, a sub-sample of the Norwegian Mother and Child cohort study (MoBa_2008, n = 1064), two Northern Finland Birth Cohort studies (NFBC1966, n = 5209, and NFBC1986, n = 2494), the Western Australian Pregnancy Cohort Study (Raine Study, n = 334), a sub-sample of Statens Serum Institut's genetic epidemiology (SSI-GE, n = 3294) studies, the Special Turku coronary Risk factor Intervention Project (STRIP, n = 441), the Diabetes and Inflammation Laboratory (1958BC (DIL-T1DGC), n = 2168) cohort, and the Wellcome Trust Case Control Consortium 1958 British Birth Cohort (1958BC (WTCCC), n = 2403). Study protocols within the EGG Consortium were approved at each study center by the local ethics committee and written informed consent had been obtained from all participants and/or their parent(s) or legal guardians. Regarding the iPSYCH and SSI-GE cohorts, GWAS data were generated based on dried blood spot samples obtained during routine neonatal screening and stored in the Danish Neonatal Screening Biobank, which is part of the Danish National Biobank. Parents are informed in writing about the neonatal screening and that the samples can later be used for research, pending approval from relevant authorities[46]. The iPSYCH and SSI-GE studies were both approved by the Danish Scientific Ethics Committees, the Danish Data Protection Agency and the Danish Neonatal Screening Biobank Steering Committee. The GPN study genotype and phenotype data were downloaded from dbGaP (https://www.ncbi.nlm.nih.gov/gap, accession number: phs000714.v1.p1) and 190 early preterm cases and 274 term infants were included in our early preterm and preterm birth meta-analysis. Study descriptions, relevant sample sizes, and basic characteristics of samples in the discovery stage are presented in Supplementary Data 1.

**Replication stage cohorts**. Three population-based cohorts with existing GWAS data were used for replication stage analyses of the lead SNPs for gestational duration and early preterm birth, namely an additional sub-sample of the Norwegian Mother and Child cohort study (MoBa_HARVEST, n = 7072), the Born in Bradford study (BiB, n = 1354), and a Finnish cohort from Helsinki (FIN, n = 865). We also included a set of mother–father–child trios from Iowa (n = 276 trios) for family-based association testing of the lead SNPs for early preterm birth. For these trios, genotyping was done using TaqMan (ThermoFisher Scientific) assays for rs1877720 (assay ID: C__12110609_10) and rs2306375 (assay ID: C__42774777_10). The study characteristics of the four replication stage cohorts are described in Supplementary Data 1.

**Exclusion criteria for cases and controls**. We excluded pregnancies based on the following criteria: (1) stillbirths; (2) twins or any multiple births; (3) ancestry outliers using principal component analysis; (4) outliers in birth weight or birth length (gestational duration possibly wrong); (5) Caesarian section, if due to pregnancy complications; Caesarian sections due to complications during labor were not excluded. Caesarian sections were allowed for cases in the postterm birth analysis; (6) physician initiated births (induced births were allowed for cases in the postterm birth analysis); (7) placental abruption, placenta previa, pre-eclampsia/eclampsia, hydramnios, placental insufficiency, cervical insufficiency, iso-immunization, gestational diabetes, cervical cerclage; (8) pre-existing medical conditions in the mother, such as diabetes, hypertension, autoimmune diseases (including systemic lupus erythematosus, rheumatoid arthritis and sclerodermia), immuno-compromised patients; and (9) known congenital anomalies. Further, the study sample was restricted to individuals of European ancestries, in most cohorts by principal component analysis. Some cohorts were not able to perform exclusions according to all criteria, but applied as many criteria as possible (see Supplementary Data 1 for details).

**Data cleaning and imputation**. Genotyping in each of the contributing studies was conducted using various high-density SNP arrays (see Supplementary Data 1 for details). Data cleaning was done locally for each study, with sample level exclusion criteria based on high genotype missing rate, high autosomal heterozygosity rate, discrepancy between reported sex and the sex inferred from genotyping, and sample heterogeneity, as well as SNP-level exclusion criteria based on call rate, Hardy–Weinberg disequilibrium, duplicate discordance, Mendelian inconsistencies, and low minor allele frequency. Imputation was performed based on reference data from the Haplotype Reference Consortium (HRC) release 1.1 (ref. [29]) for most studies. The iPSYCH sample was imputed based on the integrated phase III release of the 1000 Genomes Project[30]. Study-specific details on data cleaning filters and imputation are given in Supplementary Data 1. SNP positions were based on National Center for Biotechnology Information (NCBI) build 37 (hg19) and alleles were labeled on the positive strand of the reference genome.

**GWAS analysis**. We analyzed four traits for association with fetal genotypes, including three binary case–control traits (early preterm birth, preterm birth, and postterm birth) and one quantitative trait (gestational duration). Early preterm birth cases were defined as infants born before gestational week 34+0 (i.e. <238 days of gestation); preterm birth cases were infants born before gestational week 37+0 (i.e. <259 days of gestation, including the early cases); postterm birth cases were infants born at or after gestational week 42+0 (i.e. ≥294 days of gestation). The controls used in the analyses of these three traits were defined as infants born at or after gestational week 39+0 and before gestational week 42+0 (i.e. ≥273 days of gestation, and <294 days of gestation). Case groups in each study contributing to the discovery stage analyses were required to contain at least 50 individuals. The dichotomous trait analyses did not include additional individuals compared to the gestational duration analysis. However, these traits are of high clinical relevance and were therefore included in the study design. Also, including the dichotomous trait tests makes the study sensitive to potentially changing mechanisms influencing parturition at different pregnancy stages.

For the dichotomous traits early preterm birth, preterm birth, and postterm birth, the genome-wide association analyses within discovery studies was done by logistic regression using imputed allelic dosage data under an additive genetic model. For the quantitative trait gestational duration, we applied a rank-based inverse normal transformation. More specifically, in each cohort gestational duration in days (in some cohorts converted from weeks; see Supplementary Data 1) was regressed on infant sex and the resulting residuals were quantile transformed to a standard normal distribution before being tested for association with fetal SNP genotypes. The DNBC and MoBa_2008 samples represent case–control studies of preterm birth, which means that the distribution of gestational duration is bimodal for these studies. In these two cohorts, we transformed gestational duration to be on the same scale as the population-based cohorts (see Supplementary Methods and Supplementary Fig. 11 for details).

Some of the analyzed cohorts represent case–control studies of various diseases. For these, the association analyses of the four outcomes of interest were done in strata defined by disease group. Thus, the iPSYCH study was divided into six patient groups (autism ($n = 7147$), ADHD ($n = 8606$), schizophrenia ($n = 1101$), bipolar disorder ($n = 864$), depression ($n = 13,836$), and anorexia ($n = 1924$)) and a population control group ($n = 17,879$), which were analyzed separately and combined by fixed-effects meta-analysis. Similarly, the SSI-GE sample was split into six patient groups (atrial septal defects ($n = 368$), febrile seizures ($n = 1350$), hydrocephalus ($n = 289$), hypospadias ($n = 301$), opioid dependence ($n = 685$), and postpartum depression ($n = 301$)) (see Supplementary Data 1 for details), which were analyzed separately and combined by fixed-effects meta-analysis. Genome-wide association analyses in each cohort/sub-sample was conducted using PLINK[47], SNPTEST[48], or RVTESTS[49].

We obtained effect size estimates of the lead variant for gestational duration in the unit of days based on the iPSYCH study. Using a linear model, we regressed gestational duration in days on SNP allele dosage within each iPSYCH disease group, adjusting for infant sex. A combined estimate was obtained by fixed-effects inverse-variance meta-analysis. The iPSYCH study was also used to obtain frequency estimates for the lead variant for gestational duration within samples grouped by gestational duration. In this case, iPSYCH disease status was omitted from the model, since sample sizes would be too small if analyses were stratified by gestational duration groups as well as iPSYCH disease groups.

**Meta-analysis**. Prior to meta-analysis, SNPs with a minor allele frequency (MAF) <0.01 and poorly imputed SNPs (r2hat <0.3 from MACH[50] or info <0.4 from SNPTEST[48]) were excluded. Furthermore, SNPs available in less than 50% of the discovery cohorts for each trait were excluded. To adjust for inflation in test statistics generated in each cohort, genomic control[51] was applied once to each individual study (see Supplementary Table 6 for λ values in each study). The sub-samples within iPSYCH and SSI-GE were meta-analyzed separately first and estimates were then adjusted by genomic control again. Finally, we combined results from all discovery cohorts using fixed-effects inverse-variance-weighted meta-analysis as implemented in METAL[52]. Final meta-analysis results were obtained for 7,646,297 SNPs for gestational duration with a genomic inflation factor ($\lambda$) of 1.049, 7,588,467 SNPs for early preterm birth ($\lambda = 1.005$), 7,545,601 SNPs for preterm birth ($\lambda = 1.013$), and 7,583,965 SNPs for postterm birth ($\lambda = 1.026$). Heterogeneity between studies was estimated using the $I^2$ statistic[53]. Combined analysis of the discovery and replication stage data was also conducted by fixed-effects inverse-variance-weighted meta-analysis. We considered SNPs with $P < 0.05$ in the replication stage and $P < 5 \times 10^{-8}$ in the combined analysis to indicate robust evidence of association.

**Power analysis**. We assessed the statistical power of our study design by computer simulations in R[54]. For gestational duration, we simulated a quantitative trait influenced by an additive genetic effect and allowed the effect size and the effect allele frequency to vary. For early preterm, preterm, and postterm birth, we simulated disease state from a logistic regression model allowing the odds ratio for a log-additive genetic effect and the frequency of the effect allele to vary. For each combination of effect size and effect allele frequency, we simulated 5000 data sets using the relevant sample size (e.g., for preterm birth: 4775 cases and 60,148 controls in the discovery stage). We then conducted association tests on the

simulated data sets and calculated power as the proportion of tests with a $P$ value lower than the relevant significance level ($P < 5 \times 10^{-8}$ for the discovery stage and $P < 0.05$ for the replication stage).

**Test of non-linear effect**. At the 2q13 locus, the lead SNP was not associated with early preterm birth or preterm birth suggesting that the association with gestational duration was strongest in later stages of pregnancy. To address this question, we put forward the null hypothesis $H_0$: the variant contributes equally to higher gestational duration no matter when the child was born. We used a semi-parametric bootstrap approach to test the null hypothesis. First, we binned the 51,357 births from the largest contributing study (iPSYCH) in five groups by gestational duration. We then calculated observed allele frequencies $f1, \ldots, f5$ in the five bins. Next, we regressed gestational duration on genotype and extracted the empirical residuals. Gestational duration was now bootstrapped under the null hypothesis with resampling of the empirical residuals. Our test statistic is based on comparing allele frequencies in the five bins in 10,000 bootstrapped data sets. If the variant does not influence gestational duration as much (relative to other factors) in the early part of the distribution, then the observed allele frequency $f1$ in the first bin will be closer to the overall frequency than expected under $H_0$, while the allele frequency in the second bin ($f2$) will be lower than expected under $H_0$ and in the fifth bin ($f5$) the allele frequency will be higher than expected under $H_0$. The $P$ value for the test is calculated as the proportion of bootstrapped data sets with allele frequencies that are more extreme than the observed allele frequencies, i.e.

$$P = \frac{1}{10,000} \sum_{\text{boot}=1}^{10,000} 1(f1_{\text{boot}} > f1) * 1(f2_{\text{boot}} < f2) * 1(f5_{\text{boot}} > f5). \quad (1)$$

A more detailed description of the approach is given in the Supplementary Methods.

**Bioinformatics analysis**. To investigate the functional characteristics of our findings, we annotated all variants with $P < 1 \times 10^{-4}$ at the 2q13 locus using ANNOVAR[55] (accessed 1 June 2017), a tool that retrieves variant and region-specific functional annotations from several databases. We retrieved eQTL information for these variants from the GTEx V6 (ref. [31]) and GEUVADIS[32] project databases. We also queried GeneHancer[56], a database of human enhancers and their inferred target genes, which has integrated four different enhancer data sets, including the Encyclopedia of DNA Elements (ENCODE), the Ensembl regulatory build, the functional annotation of the mammalian genome (FANTOM) project, and the VISTA Enhancer Browser. Gene-enhancer scores (>5) were included in the annotation of the variants. We further downloaded all reported variants in the National Human Genome Research (NHGRI) GWAS Catalog[33] (accessed 24 November 2017) associated with a trait or disease at $P < 5 \times 10^{-8}$, and searched for SNPs in LD ($r^2 > 0.2$) with the lead SNP at 2q13 locus. Further annotation of these variants was performed with the Ensembl Variant Effect Predictor[57].

To assess possible enrichment of cytokine-related variants in the association results for gestational duration, we did a quantile–quantile plot of observed versus expected –log10 $P$ values of SNPs known to be associated with cytokine levels (Supplementary Fig. 9). The cytokine-related SNPs were restricted to cytokine GWAS publications[58–60], in which the association had been reported in the GWAS Catalog with $P < 5 \times 10^{-8}$.

**Exome analysis**. Exome sequencing data were available for a subset of samples in the iPSYCH study and analysis was restricted to the overlap between iPSYCH exome samples and the part of the iPSYCH cohort that were included in the GWAS. In total, $n = 18,382$ individuals, sampled from either schizophrenia ($n = 910$), bipolar ($n = 683$), ADHD ($n = 3793$), autism ($n = 5561$), affective disorder ($n = 1$), or controls ($n = 7488$) were analyzed. For these samples, variants within a 1 MB region (113–114 MB) containing the 2q13 association signal were extracted and combined with the genotype data for the lead variant rs7594852. For these variants, association analysis was performed with gestational duration, transformed as described above. We adjusted the regression model for sex and the first three principal components obtained from the genotyping data. Due to small sample size in the strata of inclusion diagnosis, we did not perform analyses within strata of inclusion diagnosis but instead performed adjustment for four indicator variables denoting whether the individual has schizophrenia, ADHD, bipolar disorder, or autism. In addition, association analysis conditioned on rs7594852 was performed by adding rs7594852 dosage as a covariate. Based on the same sequencing data, we performed a gene-based test for rare-variant association to (quantile transformed) gestational duration, using the optimal sequence kernel association test (SKAT-O)[61] approach as implemented in EPACTS version 3.2.6, using default settings.

**Estimating fetal and maternal genetic effects**. For the 2q13 locus, we analyzed 15,588 mother–child pairs from seven studies with both fetal and maternal genotypes available (ALSPAC, BiB, DNBC, EFSOCH, FIN, MoBa_2008, and MoBa_HARVEST). We used linear regression to test the association between quantile transformed gestational duration (same transformation as in the main analysis) and fetal genotype conditional on maternal genotype and vice versa. In the same complete mother–child pairs (i.e. where genotype data were available for both mother and child), we estimated unconditional effects of fetal and maternal

genotype, respectively. We combined the results from the individual studies using fixed-effects meta-analysis. To further address the question of fetal versus maternal effects, we combined unadjusted fetal effect estimates for gestational duration in days (based on 51,357 infants from the iPSYCH study) with corresponding maternal estimates from a recently published GWAS (based on 43,568 mothers)[28] using a WLM approach recently described[35]. Briefly, the fetal effect adjusted for maternal genotype is

$$\hat{\beta}_{f_{adj}} = -\frac{2}{3}\hat{\beta}_{m_{unadj}} + \frac{4}{3}\hat{\beta}_{f_{unadj}}. \quad (2)$$

And the standard error for the adjusted estimate is

$$SE\left(\hat{\beta}_{f_{adj}}\right) = \sqrt{\frac{4}{9}var\left(\hat{\beta}_{m_{unadj}}\right) + \frac{16}{9}var\left(\hat{\beta}_{f_{unadj}}\right)}. \quad (3)$$

Test statistics for the fetal adjusted effect were calculated as

$$Z_{f_{adj}} = \frac{\hat{\beta}_{f_{adj}}}{SE\left(\hat{\beta}_{f_{adj}}\right)} \quad (4)$$

and compared to a standard normal distribution to get two-sided $P$ values. Analogous formulae were used to obtain maternal results adjusted for fetal genotype. Further details and full derivations can be found in the article by Warrington et al.[35].

**Variance explained and genetic correlation analyses**. To estimate the fraction of variance in gestational duration explained by the lead variant rs7594852 at the 2q13 locus, we fitted a linear regression model of quantile transformed gestational duration in the iPSYCH cohort ($n = 51,357$) where the variant was genome-wide significant. The model was corrected for the iPSYCH disease group and the fraction of variance explained by rs7594852 genotype dosage was extracted. The estimate of variance explained by all common (MAF >1%) autosomal SNPs (also known as SNP heritability) was calculated based on the discovery stage meta-analysis results using LD Score regression[62]. The main discovery stage meta-analysis was based on quantile transformed gestational duration, but we also estimated the fraction of variance explained for gestational duration in days (based on 51,357 infants from the iPSYCH study). However, both of these estimates are influenced by fetal as well as maternal genetic loci. We therefore used the WLM approach for all common SNPs to obtain estimates of fetal effect adjusted for maternal genotype and vice versa. We then estimated the fraction of variance explained based on the WLM-adjusted results.

LD score regression[62] was used to estimate the genetic correlation between fetal effect estimates for (quantile transformed) gestational duration and effect estimates for a 690 traits and diseases in LDHub[36]. In addition to the traits available in LDHub, we calculated the genetic correlation between fetal effect estimates for gestational duration in days and corresponding estimates from a maternal GWAS of gestational duration[28], also using LD score regression.

**Computational prediction of gene regulatory mechanisms**. In order to prioritize genetic variants for experimental validation, we ranked all variants at the 2q13 locus with $r^2 > 0.8$ to the lead SNP, rs7594852, by their likelihood of being functional based on the strength of the supporting functional genomic data (e.g., ChIP-seq peaks for transcription factors or histone marks, open chromatin as measured by DNAse-seq, see Supplementary Data 7 for details). We used a wide range of functional genomic data in our analysis obtained from sources such as the UCSC Genome Browser[63], Roadmap Epigenomics[64], Cistrome[65], and ReMap-ChIP[66]. By restricting our analysis to those performed in relevant cell lines (placenta, chorion, amnion, trophoblast, neutrophils, and macrophages), we prioritized those variants likely to have regulatory function in these cells. Variants were ranked based upon the total number of data sets they overlap, which is a similar strategic scheme to that employed by RegulomeDB[67].

**Electrophoretic mobility shift assays**. EMSAs were performed to determine whether the rs7594852 polymorphism at the 2q13 locus differentially affected HIC1 binding. Recombinant human HIC1 purified protein (ORIGENE #TP322752) was obtained from ORIGENE (expressed in HEK293 using TrueORF clone, RC222752) with a c-Myc/DDK tag. Double-stranded IRDye700 5′ end-labeled 39 bp oligonucleotides, identical except for the nucleotide at rs7594852 (either the C or T allele), were obtained from IDT. The oligo sequence of the common C allele is

5′-IRDye700/
GCCAGACCCCGCCTCCTGG**C**ACAGAGGACCACGCCCGGC-3′.

The alternative T allele oligo sequence is
5′-IRDye700/
GCCAGACCCCGCCTCCTGG**T**ACAGAGGACCACGCCCGGC-3′.

The DNA-binding reaction buffer contained 1× binding buffer, 1× DTT/Tw20, 1 μg poly(dI–dC), 0.05% NP-40 (LI-COR EMSA buffer kit), and 1 mM zinc acetate. Binding reactions contained 435 ng of purified HIC1 protein. Fifty femtomoles fluorescent oligo DNAs were then added to the appropriate protein/binding mix and incubated for 20 min at room temperature. For supershift assays, 1 μg per lane at a concentration of 0.05 μg/μL of mouse anti-DDK (FLAG) monoclonal antibody

(ORIGENE #TA50011-100) was incubated with the binding buffer for 20 min prior to addition of and incubation with oligo DNA. In all, 1× orange loading dye (LI-COR kit) was added to samples, which were then resolved on (pre-cast, pre-run at 100 V for 60 min) 6% TBE gels (Novex,13 ThermoFisher) in 0.5× TBE buffer for 120 min at 80 V (4C). Fluorescent bands were then imaged using a LI-COR chemiluminescent imaging system. EMSA experiments display representative panels of 2–3 replicates. Densitometric analysis for HIC1 band intensity was performed using a Licor Odyssey scanner. The uncropped image underlying Fig. 3c is shown in the Source Data File.

**eQTL analysis in placental samples**. eQTL analyses were conducted based on existing RNA sequencing data in placental samples from the Rhode Island Child Health Study. Placenta tissues were from singleton, term pregnancies without pregnancy complications, and the original study reported eQTL results linking SNP array data with genome-wide RNA sequencing data[40]. In the eQTL analyses of the current study, the sample was restricted to 102 infants of European ancestries. Only genes/transcripts with transcription start sites within 500 kb of the lead SNP rs7594852 for gestational duration, with a total read count >50 across all samples, and with >1 counts per million (cpm) in at least two samples, were considered.

**Biomarker analysis**. Measurement of the biomarkers BDNF, CRP, EPO, IgA, IL8, IL-18, MCP1, S100B, TARC, and VEGFA was conducted based on infant dried blood spot samples obtained a few days after birth during routine neonatal screening. We tested each measured analyte for association with the lead SNP rs7594852 for gestational duration. We first fitted a linear model with age as a predictor and then normalized and log-transformed the residuals. The log-transformed residuals were tested for association with rs7594852 dosage while adjusting for infant sex, six principal components and iPSYCH disorders.

**URLs**. For 1000 Genomes Project, see http://www.1000genomes.org/; for ANNO-VAR, see http://annovar.openbioinformatics.org/; for Cis-BP, see http://cisbp.ccbr.utoronto.ca/; for dbGaP, see https://www.ncbi.nlm.nih.gov/gap; for EGG Consortium, see http://egg-consortium.org/; for Ensembl Variant Effect Predictor, see https://www.ensembl.org/vep; for EPACTS, see https://github.com/statgen/EPACTS; for GeneHancer, see http://www.genecards.org/; for GEUVADIS data browser, see http://www.ebi.ac.uk/Tools/geuvadis-das/; for GWAS Catalog, see http://www.genome.gov/gwastudies/; for Haplotype Reference Consortium, see http://www.haplotype-reference-consortium.org/; for iPSYCH, see http://ipsych.au.dk/about-ipsych/; for LDHub, see http://ldsc.broadinstitute.org/; for LD Score regression, see https://github.com/bulik/ldsc; for METAL, see http://www.sph.umich.edu/csg/abecasis/metal/; for NCBI Genotype-Tissue Expression (GTEx) eQTL database and browser, see http://www.ncbi.nlm.nih.gov/projects/gap/eqtl/index.cgi; for PLINK, see https://www.cog-genomics.org/plink2; for RVTESTS, see https://github.com/zhanxw/rvtests; for SNPTEST, see https://mathgen.stats.ox.ac.uk/genetics_software/snptest/snptest.html.

**Reporting Summary**. Further information on research design is available in the Nature Research Reporting Summary linked to this article.

## Data availability
Individual cohorts contributing to the meta-analysis should be contacted directly as each cohort has different data access policies. GWAS summary statistics from the meta-analyses of the four outcomes are available via the EGG Consortium website (https://egg-consortium.org/) and the iPSYCH website (https://ipsych.au.dk/downloads/). These summary statistics include the source data underlying Fig. 1 and Supplementary Figs. 2, 3, 4, 6 and 7. The source data underlying Figs. 2 and 3 and Supplementary Figs. 5 and 8–11 are provided as a Source Data File.

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

## Acknowledgements

We are grateful for the cooperation of the EGG Consortium and the iPSYCH-BROAD Working Group. Lists of members of the EGG Consortium and collaborators in the iPSYCH-BROAD Working Group are given in the Supplementary Notes 1 and 2, respectively. For study-specific acknowledgements, please see Supplementary Note 3. B.F. received support from an Oak Foundation fellowship, a Novo Nordisk Foundation grant (12955), and a Bill and Melinda Gates Foundation subward (137097); X.L. received support from the Nordic Center of Excellence in Health-Related e-Sciences; D.H., V.A., A.J.S., R.N., T.M.W., and A.D.B. recieved funding from the Lundbeck Foundation (R102-A9118, R155-2014-1724, R248-2017-2003); L.S. reports funding from a Carlsberg Foundation postdoctoral fellowship (CF15-0899); R.M.F. is a Sir Henry Dale Fellow (Wellcome and Royal Society grant: WT104150); R.N.B. is funded by Wellcome and Royal Society (grant: WT104150); M.C.B. and D.A.L. work in a unit that receives UK MRC funding (MC_UU_00011/6); M.C.B. is supported by the MRC Skills Development Fellowship MR/P014054/1; D.A.L.'s contribution to this work was funded by grants from the US NIH and European Research Council under the European Union's Seventh Framework Programme (FP/2007–2013)/ERC Grant Agreement (Grant number 669545; DevelopObese) and European Union's Horizon 2020 research and innovation programme under grant agreement 733206 (LIFECYCLE); D.A.L. is also an NIHR senior investigator (NF-SI-0611-10,196); F.R. received partial funding from the Netherlands Organization for Health Research and Development (VIDI 016.136.367); J.F.F. and V.W.V.J. received partial funding from the European Union's Horizon 2020 research and innovation programme under grant agreements 733206 (LIFECYCLE) and 633595 (DynaHEALTH); V.W.V.J. received partial funding from the Netherlands Organization for Health Research and Development (VIDI 016.136.361) and the European Research Council (ERC Consolidator Grant, ERC-2014-CoG-648916); I.K. reports funding from Sigrid Juselius Foundation and Biomendicum Foundation postdoctoral fellowship; L.J.M. was supported by the March of Dimes Prematurity Research Center Ohio Collaborative, NICHD HD 091527, and the Bill and Melinda Gates Foundation (OPP1113966); M.T.W. was supported by NIH R01 NS099068, NIH R01 GM055479-19A1, a Lupus Research Alliance Novel Approaches award, CCRF Endowed Scholar, a CCHMC CpG Pilot study award, and a CCHMC Trustee award; K.K.R. received support from March of Dimes (21-FY13-19); C.P. was supported by the NIHR Great Ormond Street Hospital Biomedical Research Centre; M.I.M. is a Wellcome Senior Investigator and a NIHR Senior Investigator. His work is supported by Wellcome (090532, 093831, 203141, 106130) and by the NIH (U01DK105535). The views expressed in this article are those of the authors and not necessarily those of the NHS, the NIHR, or the Department of Health. For study-specific funding, please see Supplementary Note 4.

## Author contributions

Statistical analysis: X.L., D.H., L.S., R.N.B., M.W., F.G., J.J., A.M., J.P.B., F.T.J.L., S.V., T.S.A., N.P., C.A.W., M.C.B., G.Z., I.K., M.G.H., D.M.S., J.F., R.N., L.-P.L., J.F.F., E.H., R.M.F., A.B., and B.F. Study design: D.A.L., C.P., M.I.M., H.A.B., M.L.M., H.H., V.W.V.J., R.K.V., H.B., K.P., O.R., C.E.P., K.B., J.F.F., S.F.A.G., E.H., T.M.W., M.M., A.B., and B.F. Sample collection: D.A.L., C.P., H.A.B., H.H., V.W.V.J., R.K.V., H.B., B.A.K., K.P., O.R., D.M.H., C.E.P., K.B., M.V., J.F.F., W.L.L., S.F.A.G., E.H., M.-R.J., J.C.M., M.M., and B.F. Genotyping: X.L., F.G., J.J., A.M., M.B., J.B., B.A.B., W.K.T., V.A., D.A.L., M.I.M., M.L.M., H.H., M.G.H., F.R., H.B., K.P., O.R., Ø.H., S.J., P.R.N., A.J.S., P.B.M., A.D.B., M.N., O.M., K.K.R., D.M.H., C.E.P., K.B., W.L.L., S.F.A.G., B.J., M.-R.J., L.J.M., J.C.M., R.M.F., T.M.W., and A.B. Functional follow-up experiments: R.M.R., K.S., S.P., D.E.M., X.C., M.T.W., K.H., D.M.H., L.C.K., L.J.M. Writing and overall study direction: X.L., D.H., L.S., J.C.M., L.J.M., R.M.F., T.M.W., M.M., A.B., and B.F. All authors reviewed and edited the manuscript.

## Additional information

**Competing interests:** D.A.L. received support from Roche Diagnostics and Medtronic for biomarker research unrelated to the work presented in this paper. M.I.M. serves on advisory panels for Pfizer, NovoNordisk, Zoe Global; has received honoraria from Merck, Pfizer, NovoNordisk, and Eli Lilly; has stock options in Zoe Global; has received research funding from Abbvie, Astra Zeneca, Boehringer Ingelheim, Eli Lilly, Janssen, Merck, NovoNordisk, Pfizer, Roche, Sanofi Aventis, Servier & Takeda. S.F.A.G. has received support from GSK for research that is not related to the study presented in this paper. T.M.W. has acted as lecturer and scientific advisor to H. Lundbeck A/S. The remaining authors declare no competing interests.

Xueping Liu[1,71], Dorte Helenius[2,3,71], Line Skotte[1,71], Robin N. Beaumont[4], Matthias Wielscher[5], Frank Geller[1], Julius Juodakis[6], Anubha Mahajan[7], Jonathan P. Bradfield[8,9], Frederick T.J. Lin[10], Suzanne Vogelezang[11,12,13], Mariona Bustamante[14,15,16], Tarunveer S. Ahluwalia[17], Niina Pitkänen[18], Carol A. Wang[19], Jonas Bacelis[20], Maria C. Borges[21,22], Ge Zhang[23,24,25], Bruce A. Bedell[26], Robert M. Rossi[25,27], Kristin Skogstrand[2,28], Shouneng Peng[29,30], Wesley K. Thompson[2,3], Vivek Appadurai[2,3], Debbie A. Lawlor[21,22,31], Ilkka Kalliala[32,33], Christine Power[34], Mark I. McCarthy[7,35,36], Heather A. Boyd[1], Mary L. Marazita[37,38], Hakon Hakonarson[8,39,40], M. Geoffrey Hayes[10,41,42], Denise M. Scholtens[43], Fernando Rivadeneira[11,13,44], Vincent W.V. Jaddoe[11,12,13], Rebecca K. Vinding[17], Hans Bisgaard[17], Bridget A. Knight[45], Katja Pahkala[18,46], Olli Raitakari[18,47], Øyvind Helgeland[48,49,50], Stefan Johansson[48,51], Pål R. Njølstad[48,49], João Fadista[1,52], Andrew J. Schork[2,3], Ron Nudel[2,3], Daniel E. Miller[53], Xiaoting Chen[53], Matthew T. Weirauch[27,53,54], Preben Bo Mortensen[2,55,56], Anders D. Børglum[2,56,57], Merete Nordentoft[2,58,59], Ole Mors[2,60], Ke Hao[29,30],

Kelli K. Ryckman[26,61], David M. Hougaard [2,28], Leah C. Kottyan [27,53], Craig E. Pennell[19], Leo-Pekka Lyytikainen [62,63], Klaus Bønnelykke [17], Martine Vrijheid[14,15,16], Janine F. Felix [11,12,13], William L. Lowe Jr[10], Struan F.A. Grant [8,39,40], Elina Hyppönen [34,64,65], Bo Jacobsson [6,66], Marjo-Riitta Jarvelin [67,68], Louis J. Muglia [23,24,26,27], Jeffrey C. Murray[26], Rachel M. Freathy [4,69], Thomas M. Werge [2,3,59], Mads Melbye [1,59,70], Alfonso Buil [2,3,72] & Bjarke Feenstra [1,72]

[1]Department of Epidemiology Research, Statens Serum Institut, Copenhagen, Denmark. [2]iPSYCH, The Lundbeck Foundation Initiative for Integrative Psychiatric Research, Aarhus, Denmark. [3]Institute of Biological Psychiatry, Mental Health Center Sct. Hans, Mental Health Services Copenhagen, Roskilde, Denmark. [4]Institute of Biomedical and Clinical Science, College of Medicine and Health, University of Exeter Medical School, University of Exeter, Royal Devon and Exeter Hospital, Barrack Road, Exeter EX2 5DW, UK. [5]Department of Epidemiology and Biostatistics, MRC-PHE Centre for Environment and Health, School of Public Health, Imperial College London, London, UK. [6]Department of Obstetrics and Gynecology, Sahlgrenska Academy, University of Gothenburg, Gothenburg, Sweden. [7]Wellcome Centre for Human Genetics, University of Oxford, Oxford OX3 7BN, UK. [8]Center for Applied Genomics, The Children's Hospital of Philadelphia, Philadelphia, PA, USA. [9]Quantinuum Research, LLC, San Diego, CA, USA. [10]Division of Endocrinology, Metabolism and Molecular Medicine, Department of Medicine, Northwestern University Feinberg School of Medicine, Chicago, IL, USA. [11]The Generation R Study Group, Erasmus MC, University Medical Center Rotterdam, Rotterdam, The Netherlands. [12]Department of Pediatrics, Erasmus MC, University Medical Center Rotterdam, Rotterdam, The Netherlands. [13]Department of Epidemiology, Erasmus MC, University Medical Center Rotterdam, Rotterdam, The Netherlands. [14]ISGlobal, Barcelona Institute for Global Health, Barcelona, Spain. [15]Universitat Pompeu Fabra (UPF), Barcelona, Spain. [16]CIBER Epidemiología y Salud Pública (CIBERESP), Madrid, Spain. [17]COPSAC, Copenhagen Prospective Studies on Asthma in Childhood, Herlev and Gentofte Hospital, University of Copenhagen, Copenhagen, Denmark. [18]Research Centre of Applied and Preventive Cardiovascular Medicine, University of Turku, 20520 Turku, Finland. [19]School of Medicine and Public Health, Faculty of Medicine and Health, The University of Newcastle, Newcastle, NSW, Australia. [20]Department of Obstetrics and Gynecology, Sahlgrenska University Hospital, Gothenburg, Sweden. [21]MRC Integrative Epidemiology Unit at the University of Bristol, Bristol, UK. [22]Population Health Sciences, Bristol Medical School, University of Bristol, Bristol, UK. [23]Division of Human Genetics, Cincinnati Children's Hospital Medical Center, Cincinnati, OH, USA. [24]Center for Prevention of Preterm Birth, Perinatal Institute, Cincinnati Children's Hospital Medical Center, Cincinnati, OH, USA. [25]March of Dimes Prematurity Research Center Ohio Collaborative, Cincinnati, OH, USA. [26]Department of Pediatrics, University of Iowa, Iowa City, IA, USA. [27]Department of Pediatrics, University of Cincinnati College of Medicine, Cincinnati, OH, USA. [28]Statens Serum Institut, Center for Neonatal Screening, Department for Congenital Disorders, Copenhagen, Denmark. [29]Department of Genetics and Genomic Sciences, Icahn School of Medicine at Mount Sinai, 1425 Madison Avenue, New York, NY 10029, USA. [30]Icahn Institute of Genomics and Multiscale Biology, Icahn School of Medicine at Mount Sinai, 1425 Madison Avenue, New York, NY 10029, USA. [31]NIHR Bristol Biomedical Research Centre, Bristol, UK. [32]Department of Surgery and Cancer, IRDB, Faculty of Medicine, Imperial College, London W12 0NN, UK. [33]Department of Obstetrics and Gynaecology, University of Helsinki and Helsinki University Hospital, Haartmaninkatu 2 00029 HUS, Finland. [34]Population, Policy and Practice, Great Ormond Street Institute for Child Health, University College London, London, UK. [35]Oxford Centre for Diabetes, Endocrinology and Metabolism, University of Oxford, Oxford OX3 7LJ, UK. [36]NIHR Oxford Biomedical Research Centre, Churchill Hospital, Oxford OX3 7LJ, UK. [37]Center for Craniofacial and Dental Genetics, Department of Oral Biology School of Dental Medicine, University of Pittsburgh, Pittsburgh, PA, USA. [38]Department of Human Genetics, Graduate School of Public Health, University of Pittsburgh, Pittsburgh, PA, USA. [39]Division of Human Genetics, The Children's Hospital of Philadelphia, Philadelphia, PA, USA. [40]Department of Pediatrics, Perelman School of Medicine, University of Pennsylvania, Philadelphia, PA, USA. [41]Department of Anthropology, Northwestern University, Evanston, IL, USA. [42]Center for Genetic Medicine, Northwestern University Feinberg School of Medicine, Chicago, IL, USA. [43]Division of Biostatistics, Department of Preventive Medicine, Northwestern University Feinberg School of Medicine, Chicago, IL, USA. [44]Department of Internal Medicine, Erasmus MC, University Medical Center Rotterdam, Rotterdam, The Netherlands. [45]NIHR Exeter Clinical Research Facility, University of Exeter Medical School, University of Exeter, Royal Devon and Exeter Hospital, Barrack Road, Exeter EX2 5DW, UK. [46]Paavo Nurmi Centre, Sports & Exercise Medicine Unit, Department of Health and Physical Activity, University of Turku, 20520 Turku, Finland. [47]Department of Clinical Physiology and Nuclear Medicine, Turku University Hospital, 20521 Turku, Finland. [48]K. G. Jebsen Center for Diabetes Research, Department of Clinical Science, University of Bergen, Bergen, Norway. [49]Department of Pediatrics, Haukeland University Hospital, Bergen, Norway. [50]Norwegian Institute of Public Health, Division of Health data and Digitalization, Department of Genetic Research and Bioinformatics, Oslo, Norway. [51]Center for Medical Genetics and Molecular Medicine, Haukeland University Hospital, Bergen, Norway. [52]Lund University Diabetes Centre, Department of Clinical Sciences, Lund University, Malmö, Sweden. [53]Center for Autoimmune Genomics and Etiology, Cincinnati Children's Hospital Medical Center, Cincinnati, OH, USA. [54]Divisions of Biomedical Informatics and Developmental Biology, Cincinnati Children's Hospital, Cincinnati, OH, USA. [55]National Centre for Register-Based Research, Aarhus University, Aarhus, Denmark. [56]iSEQ, Centre for Integrative Sequencing, Aarhus University, Aarhus, Denmark. [57]Department of Biomedicine—Human Genetics, Aarhus University, Aarhus, Denmark. [58]Mental Health Center Copenhagen, Mental Health Services in the Capital Region of Denmark, Copenhagen, Denmark. [59]Department of Clinical Medicine, Faculty of Health and Medical Sciences, University of Copenhagen, Copenhagen, Denmark. [60]Psychosis Research Unit, Aarhus University Hospital, Risskov, Denmark. [61]Department of Epidemiology, University of Iowa, Iowa City, IA, USA. [62]Department of Clinical Chemistry, Fimlab Laboratories, Tampere 33520, Finland. [63]Department of Clinical Chemistry, Finnish Cardiovascular Research Center—Tampere, Faculty of Medicine and Life Sciences, University of Tampere, 33014 Tampere, Finland. [64]Australian Centre for Precision Health, University of South Australia Cancer Research Institute, Adelaide, Australia. [65]South Australian Health and Medical Research Institute, Adelaide, Australia. [66]Department of Genes and Environment, Division of Epidemiology, Norwegian Institute of Public Health, Oslo, Norway. [67]Institute of Health Sciences, University of Oulu, Oulu, Finland. [68]Department of Epidemiology and Biostatistics, School of Public Health, Medical Research Council-Health Protection Agency Centre for Environment and Health, Faculty of Medicine, Imperial College London, London, UK. [69]Medical Research Council Integrative Epidemiology Unit at the University of Bristol, Oakfield House, Oakfield Grove, Bristol BS8 2BN, UK. [70]Department of Medicine, Stanford University School of Medicine, Stanford, CA, USA. [71]These authors contributed equally: Xueping Liu, Dorte Helenius, Line Skotte. [72]These authors jointly supervised this work: Alfonso Buil, Bjarke Feenstra.

