## [Peer Review File · Nature Communications]

Reviewers' Comments:

Reviewer #1:

Remarks to the Author:

The authors performed an genome-wide meta-analysis of gestational age at birth on fetal DNA samples totaling over 84k subjects. A single locus was identified and later confirmed in about 9k additional infants and was distinguished from the influence of maternal genotype in about 15K maternal-child dyads. The lead SNP is related to inflammatory pathways which are relevant for the timing of birth. The application of statistical-genetic methods in this study was rigorous and makes a strong case for the association of a fetal SNP with gestational age at birth (GA). A sufficiently sized replication sample for GA confirms their lead finding. Follow-up of association results in maternal-child dyads provided sufficient support for a fetal versus maternal SNP effect.

While this study is viewed as excellent and well written, I do have a few comments that I hope will make it accessible to a wider range of readers:

page 10, line 101: The test used that concludes 7.3% of variance by all common SNPs should be indicated in the Methods. Additionally, this observation is striking and in my mind is the most exciting result from this study. Being that studies have shown 10-20% of variance in GA is estimated from the fetal component one would conclude this study has found direct evidence for about one-half. M-GCTA studies could further confirm the fetal versus maternal contribution.

page 10, line 103: The statement that rs7594852 influences GA at later stages of pregnancy does not reconcile with the assumptions of the biometrical genetic model. The C allele contributes to a higher GA conditioned on an individual's genetic background (no matter when the child was born). The statistical-genetic model used does not indicate an effect of the SNP at any specific stage of GA (early or late). It is not clear that Supp Fig 4 means what the authors think it does. In this regard, the Discussion contains several instances of inconsistent reasoning.

It is not clear how dichotomizing GA into early PTB, PTB and post-term phenotypes reconciles with the additive genetic tests performed, being that there is no biometrical genetic model that maps additive genetic variation on a truly dichotomous trait. The assumption of the biometrical model is that all 3 binary traits map to the same underlying (continuous) risk liability which is best represented by the quantitative gestation duration phenotype. Thus, a finding in one phenotype would have the same interpretation across all phenotypes. Thus, removing the dichotomous trait analysis would improve the clarity of this study without any loss of information.

Reviewer #2:

Remarks to the Author:

Liu et al. conducted GWA meta-analysis of gestational duration and early preterm, preterm and postterm birth in over 84K infants and identified a locus on chromosome 2q13, which was independently replicated in additional ~9K infants. This is an interesting and well conducted study with important findings, but readers will benefit if the following are also addressed:

1. Authors analyzed whole-exome sequencing data with respect to the 2q13 association signal but only examined single variant at a time. It would be important to examine coding variants jointly using burden and/or SKAT tests, if possible, exome-wide given iPSYCH data is available.
2. Four traits analyzed here may be correlated and may have some common genetic background. Could authors estimate genetic correlation among these and probably also perform meta-analysis for 4 traits? If there are common genetic signals, they may be discovered in the combined analysis but may be weaker when examining separately.
3. Also, consider examining the results using LD-hub to see if there are any genetically related traits (gynecological etc.), especially given a endometriosis association at the 2q13 locus which

was moderately correlated with the top SNP reported here.

Reviewer #3:

Remarks to the Author:

1. This paper examines GWAS for essentially one outcome, gestational age, but modelled in 3 ways, continuous and 2 types of categorical extreme groups. The findings are incongruent between these different models and more rationale is needed as to why these separate models are needed and are valid, rather than simply increasing the risk of false positives due to multiple slicing of one dataset.
2. Comparison of fetal and maternal associations in conditional models is a key part of the paper (results on Lines 167-168). But there does not appear to be a formal test to indicate if a SNP is maternal or fetal or both? Instead the authors rely on a visual inspection of effect sizes. Results for the other 6 (4) published maternal GWAS SNPs should be described in more detail in the text.
3. With only one confirmed GWAS hit identified, robust insights for this 2q13 locus are essential. However the functional assays for 2q13, which lies in a dense gene region, are very limited for the following reasons i) the selection of the top SNP at this locus for functional testing is based on a metric that favours wide pleiotropic actions (L175) rather than a specific function which is also plausible; ii) they overlook the 4 missense variants identified in the discovery sample because these were not significant in their much smaller follow-up exome sequencing in $n=11,455$, iii) very limited sample size for studies of HIC1 binding ($n=4$, $P=0.013$) and lack of a functional connection between HIC 1 and their placental expression data.
4. The authors should report more comprehensively in the text the associations of all highlighted GWAS SNPs on continuous GA, preterm and early preterm lead SNPs.
5. Several of the Discovery cohorts may have excluded or undersampled preterm and early preterm infants. Comment on how the frequency of preterm ($n=4775$) and early preterm ($n=1139$) births in the overall sample ($n=84,689$) compares to an unselected sample.
6. What was the estimated heritability of gestational age, preterm and early preterm birth based on GWAS SNPs? Only 1 significant SNP in a sample of 80,000 indicates that this trait has extremely low heritability.
7. They explore the seeming non-linear effect of the 2q13 signal on gestational age on pages 103-114. The results are based on a simple visual impression. This needs formal testing for non-linear association.
8. Line 118 - Report in the text which genes harbour the 4 missense SNPs at 2q13.
9. Line 161-163 - Results of fetal-maternal conditional models are key to this paper and should be reported in the text or main display rather than just citing the supplementary data.

NCOMMS-18-30828-T

Point-by-point response to reviewers

We thank the reviewers for the positive feedback and for their helpful comments and suggestions, which have helped us improve the manuscript considerably.

Reviewers' comments:

Reviewer #1 (Remarks to the Author):

The authors performed an genome-wide meta-analysis of gestational age at birth on fetal DNA samples totaling over 84k subjects. A single locus was identified and later confirmed in about 9k additional infants and was distinguished from the influence of maternal genotype in about 15K maternal-child dyads. The lead SNP is related to inflammatory pathways which are relevant for the timing of birth. The application of statistical-genetic methods in this study was rigorous and makes a strong case for the association of a fetal SNP with gestational age at birth (GA). A sufficiently sized replication sample for GA confirms their lead finding. Follow-up of association results in maternal-child dyads provided sufficient support for a fetal versus maternal SNP effect.

While this study is viewed as excellent and well written, I do have a few comments that I hope will make it accessible to a wider range of readers:

page 10, line 101: The test used that concludes 7.3% of variance by all common SNPs should be indicated in the Methods. Additionally, this observation is striking and in my mind is the most exciting result from this study. Being that studies have shown 10-20% of variance in GA is estimated from the fetal component one would conclude this study has found direct evidence for about one-half. M-GCTA studies could further confirm the fetal versus maternal contribution.

The primary objective of our study was to detect robustly associated loci in the fetal genome and to investigate possible molecular mechanisms underlying the genetic associations. However, we agree with the reviewer that understanding how much of the variance in gestational duration can be attributed to fetal genetic variation is an important question, which we have now treated in greater depth. First, it is worth noting that our main discovery meta-analysis was based on quantile transformed gestational duration. Analyzing untransformed gestational duration in days (based on 51,357 infants from the iPSYCH study) gave a lower estimate of 4.5% variance explained. Importantly, however, both of these estimates do not take into account the influence of maternal genetics. We therefore used a recently developed weighted linear model (WLM) approach (Warrington et al. 2019 Nature Genetics (in press); preprint: <https://doi.org/10.1101/442756>) for all common SNPs to obtain estimates of fetal effect adjusted for maternal genotype and vice versa. Based on these WLM-adjusted estimates, we used LD score regression to estimate the proportion of variance explained by common autosomal fetal and maternal variants. The proportion of variance explained by WLM-adjusted fetal effects was 1.3% (SE=1.0%) and that of WLM-adjusted maternal effects was 4.9% (SE=1.3%), which is lower than corresponding estimates from family-based studies. This phenomenon of missing heritability has been observed for many traits and diseases, and we discuss possible underlying factors.

The Methods section has been updated to clarify how we estimated variance explained, and the following new paragraphs were added to the Results and Discussion sections of the manuscript.

New paragraph in the Results section:

“Heritability and genetic correlation with other traits

Based on the gestational duration summary statistics for all common autosomal SNPs (minor allele frequency, MAF>1%), the estimated proportion of variance explained (SNP heritability) was 7.6% (SE=0.8%). This estimate was based on the quantile transformed phenotype, and when using results for gestational duration in days (based on 51,357 infants from the iPSYCH study) the variance explained was 4.5% (SE=1.1%). For comparison, we analyzed summary statistics from a recent maternal GWAS of gestational duration in days (based on 43,568 mothers)²⁷ using the same SNP set and found that the proportion of variance explained was 7.9% (SE=1.5%). However, the above estimates are all influenced by fetal as well as maternal genetic loci. To obtain estimates of fetal effect adjusted for the maternal genotype and vice versa for each SNP, we combined the unadjusted fetal effects with unadjusted maternal effects (based on gestational duration in days) using the WLM approach³⁵. The proportion of variance explained by WLM-adjusted fetal effects was 1.3% (SE=1.0%) and that of WLM-adjusted maternal effects was 4.9% (SE=1.3%).

As expected there was a strong positive correlation between unadjusted fetal and maternal effects for gestational duration in days ($r_g = 0.77$, SE = 0.17, $P = 4.29 \times 10^{-6}$). When performing genetic correlation analyses between our meta-analysis results for (quantile transformed) gestational duration and 690 traits and diseases in LDHub³⁴, we found that eight were significant after correction for multiple testing (**Supplementary Table 9**). These included positive genetic correlations with own birth weight ($r_g = 0.21$, SE = 0.04, $P = 4.14 \times 10^{-6}$) and birth weight of first child ($r_g = 0.28$, SE = 0.05, $P = 1.23 \times 10^{-8}$).

New paragraph in the Discussion:

“Looking across the genome, we found that common autosomal fetal genetic variants explained 7.6% of the variance in (quantile transformed) gestational duration. When instead analyzing gestational duration in days (untransformed, based on 51,357 infants from the iPSYCH study), the fraction of variance explained by common fetal variants was 4.5%. However, to fully address the question of variance explained by fetal genetic variation, the maternal genetic contribution needs to be accounted for. Combining fetal results for gestational duration in days with corresponding maternal results from an independent sample²⁸ using the WLM approach, the fraction of variance explained was 1.3% for WLM-adjusted fetal effects and 4.9% for WLM-adjusted maternal effects. The larger influence of maternal compared to fetal genetic variation on gestational duration is consistent with findings from large family studies in populations of Scandinavian origin^{21,22}, but our estimates of variance explained are lower, both before and after WLM adjustment. Such missing heritability has been observed for many traits and diseases, and is often attributed to rare causal variants in low LD with common SNPs as well as possible overestimation of heritability in family studies due to shared environmental effects or non-additive genetic variation⁴¹. Furthermore, we note that giving more weight to observations in the lower tail of the distribution (by going from the quantile transformed phenotype to the untransformed phenotype) resulted in lower heritability estimates. This was also observed in a large family study, which therefore excluded births before week 35 when estimating heritability²¹. A detailed dissection of fetal and maternal contributions to the heritability of gestational duration lies beyond the scope of the current study, but is an important topic for future research.”

page 10, line 103: The statement that rs7594852 influences GA at later stages of pregnancy does not reconcile with the assumptions of the biometrical genetic model. The C allele contributes to a higher GA conditioned on an individual's genetic background (no matter when the child was born). The statistical-

genetic model used does not indicate an effect of the SNP at any specific stage of GA (early or late). It is not clear that Supp Fig 4 means what the authors think it does. In this regard, the Discussion contains several instances of inconsistent reasoning.

We agree with the reviewer that the statistical-genetic model applied in the genome-wide scan to identify gestational duration associated loci does assume a general shift in the whole distribution of gestational duration conditioned on the genotype. However, in practice this assumption may not be entirely true, e.g., in the case of a phenotype of heterogeneous etiology like gestational duration, where one set of mechanisms may be playing a role in causing early parturition before the mechanisms that would typically initialize parturition around term get their chance to influence the phenotype.

We have addressed the question of possible differences in the strength of the genetic association over the course of gestation in much greater detail and have replaced the previous figure with a new, more easily interpretable one (Supplementary Fig. 5). More specifically, we carried out semi-parametric bootstrapping under the null hypothesis assumption H_0 : "the variant contributes equally to higher gestational duration no matter when the child was born". We chose a semi-parametric bootstrap approach to avoid assuming normal-distributed residuals in the distribution of gestational duration. A detailed description is given in the Supplementary Methods, but we outline the statistical test here. Note that we decided to divide the data into 5 bins instead of the previous 10 bins, to increase the statistical power by increasing the number of observations in each bin. This is preferable because of the modest effect size of the genetic variant.

Our test statistic is based on bootstrapping allele frequencies in the five bins under H_0 . If the variant does not influence gestational duration as much (relative to other factors) in the early part of the distribution, then the observed allele frequency f_1 in the first bin will be closer to the overall frequency than expected under H_0 , while the allele frequency in the second bin (f_2) will be lower than expected under H_0 and in the fifth bin (f_5) the allele frequency will be higher than expected under H_0 . The semi-parametric bootstrapping under H_0 was based on imputed genotype dosages and gestational duration in days in the iPSYCH sample. Based on 10,000 joint bootstrap distributions of gestational duration and genotype, we estimate that the probability under H_0 of observing more deviating allele frequencies is $P = 0.0013$. We also provide a figure (Supplementary Fig. 5) illustrating the expected binned allele frequencies under H_0 obtained under the bootstrap procedure.

To clarify these points in the main text of the manuscript, we have rephrased the relevant paragraph in the Results section from:

"No association was seen at the 2q13 locus in case-control analyses of early preterm birth or preterm birth (Supplementary Fig. 4), suggesting that the locus primarily influences gestational duration in the later stages of pregnancy. To further investigate this question, we binned the 51,357 births from the largest contributing study (iPSYCH) in 10 groups of equal size by gestational duration. We then estimated the frequency of the rs7594852-C allele in each group and in the whole sample. In the overall meta-analysis, each additional fetal rs7594852-C allele was associated with increased gestational duration (Table 1). The frequencies of the rs7594852-C allele in the two groups with shortest gestational durations were only slightly lower than the frequency in the whole sample (Supplementary Fig. 5). The lowest allele frequency (0.514) was seen in the third group, representing a mean gestational duration of 276 days. The allele frequency then gradually increased in the next groups with the highest frequency (0.551) observed for the group representing the longest gestational duration (mean of 296 days) (Supplementary Fig. 5)."

to:

"No association was seen at the 2q13 locus in case-control analyses of early preterm birth or preterm birth (Supplementary Fig. 4). This may suggest that other mechanisms could be playing a role in causing early parturition before the mechanisms mediating the effect of the locus get the opportunity to influence the phenotype. To further investigate this question, we binned the 51,357 births from the largest contributing study (iPSYCH) in 5 groups by gestational duration. We then estimated the frequency of the rs7594852-C allele in each group and in the whole sample. In the overall meta-analysis, each additional fetal rs7594852-C allele was associated with increased gestational duration (Table 1). The frequency of the rs7594852-C allele in the group with the shortest gestational duration was only slightly lower than the frequency in the whole sample (Supplementary Fig. 5). The lowest allele frequency (0.518) was seen in the second group, representing a mean gestational duration of 276.5 days. The allele frequency then gradually increased in the next groups with the highest frequency (0.555) observed for the group representing the longest gestational duration (mean of 298.3 days) (Supplementary Fig. 5). This pattern in allele frequencies deviates from what is expected under the hypothesis that the strength of the association is independent of gestational duration ($P = 0.0013$, semi-parametric bootstrap, see Supplementary Methods for details)."

It is not clear how dichotomizing GA into early PTB, PTB and post-term phenotypes reconciles with the additive genetic tests performed, being that there is no biometrical genetic model that maps additive genetic variation on a truly dichotomous trait. The assumption of the biometrical model is that all 3 binary traits map to the same underlying (continuous) risk liability which is best represented by the quantitative gestation duration phenotype. Thus, a finding in one phenotype would have the same interpretation across all phenotypes. Thus, removing the dichotomous trait analysis would improve the clarity of this study without any loss of information.

If duration of gestation was a "pure" trait we would agree with the reviewer that early PTB, PTB and postterm birth merely represent dichotomizations of the same underlying liability scale. However, we regard gestational duration as a phenotype of heterogeneous etiology in the sense that some mechanisms may cause early parturition with a timing that is independent of the set of mechanisms that trigger parturition when gestation is complete. We therefore designed our study to use tests that are sensitive to potential different mechanisms at different pregnancy stages by analyzing the quantitative trait and the three dichotomous traits. Furthermore, we believe that results for the clinically relevant dichotomous traits will be of interest and expected among clinically interested readers of the article.

To clarify the rationale for analyzing the quantitative trait of gestational duration as well as the three dichotomous traits, we now emphasize in the Introduction that these are clinically relevant, and include the following in the Methods section:

"The dichotomous trait analyses did not include additional individuals compared to the gestational duration analysis. However, these traits are of high clinical relevance and were therefore included in the study design. Also, including the dichotomous trait tests makes the study sensitive to potentially changing mechanisms influencing parturition at different pregnancy stages."

Reviewer #2 (Remarks to the Author):

Liu et al. conducted GWA meta-analysis of gestational duration and early preterm, preterm and postterm birth in over 84K infants and identified a locus on chromosome 2q13, which was independently

replicated in additional ~9K infants. This is an interesting and well conducted study with important findings, but readers will benefit if the following are also addressed:

1. Authors analyzed whole-exome sequencing data with respect to the 2q13 association signal but only examined single variant at a time. It would be important to examine coding variants jointly using burden and/or SKAT tests, if possible, exome-wide given iPSYCH data is available.

Since the first submission of the manuscript, exome sequencing data have been generated for an additional set of iPSYCH study participants increasing the total sample to 18,382. Using the larger data set, we have updated the single variant analyses and have also followed the reviewer's suggestion and conducted joint tests of coding variants in the genes at the 2q13 locus using the optimal sequence kernel association test (SKAT-O) approach (Lee et al (2012); PMID: 22699862). These analyses did not suggest that exonic variants could explain the observed association at the 2q13 locus. The design of our study was based on meta-analysis of GWAS data sets in the discovery stage, but we hope to be able to conduct an exome-wide study as larger sample sets with exome sequencing data and information on gestational duration become available.

The relevant paragraphs in the Methods and Results sections have been updated with descriptions and results from the SKAT-O analyses.

2. Four traits analyzed here may be correlated and may have some common genetic background. Could authors estimate genetic correlation among these and probably also perform meta-analysis for 4 traits? If there are common genetic signals, they may be discovered in the combined analysis but may be weaker when examining separately.

We appreciate the suggestion of gaining statistical power by meta-analysis of traits that are highly correlated. However, the individuals contributing to the discovery analyses for the dichotomous traits were subsets of the individuals contributing to the gestational duration discovery analysis. The rationale for performing analyses for these clinically relevant dichotomous outcomes was to be sensitive to scenarios where, for example, some mechanisms may cause early parturition, with a timing that is independent of the set of mechanisms that trigger parturition when gestation is complete (also outlined in the response to reviewer 1, comment #3). We have clarified these points with the following sentences in the Methods section:

“The dichotomous trait analyses did not include additional individuals compared to the gestational duration analysis. However, these traits are of high clinical relevance and were therefore included in the study design. Also, including the dichotomous trait tests makes the study sensitive to potentially changing mechanisms influencing parturition at different pregnancy stages.”

3. Also, consider examining the results using LD-hub to see if there are any genetically related traits (gynecological etc.), especially given a endometriosis association at the 2q13 locus which was moderately correlated with the top SNP reported here.

We have followed the reviewer's suggestion, and estimated genetic correlation between our main meta-analysis results for quantile transformed gestational duration and 690 traits and diseases in LDHub. Endometriosis was however not available. We further calculated the genetic correlation between fetal and maternal effects for gestational duration in days. The following has been added to the Results section:

“As expected there was a strong positive correlation between unadjusted fetal and maternal effects for gestational duration in days ($r_g = 0.77$, $SE = 0.17$, $P = 4.29 \times 10^{-6}$). When performing genetic correlation analyses between our meta-analysis results for (quantile transformed) gestational duration and 690 traits and diseases in LDHub³⁴, we found that eight were significant after correction for multiple testing (Supplementary Table 9). These included positive genetic correlations with own birth weight ($r_g = 0.21$, $SE = 0.04$, $P = 4.14 \times 10^{-6}$) and birth weight of first child ($r_g = 0.28$, $SE = 0.05$, $P = 1.23 \times 10^{-8}$).”

Reviewer #3 (Remarks to the Author):

1. This paper examines GWAS for essentially one outcome, gestational age, but modelled in 3 ways, continuous and 2 types of categorical extreme groups. The findings are incongruent between these different models and more rationale is needed as to why these separate models are needed and are valid, rather than simply increasing the risk of false positives due to multiple slicing of one dataset.

While gestational duration at parturition is an easily defined quantitative trait, we are not able to rule out the possibility that it represents a phenotype of heterogeneous etiology, such that some mechanisms may cause early parturition, with a timing that is independent of the set of mechanisms that trigger parturition when gestation is complete (also outlined in the response to reviewer 1, comment #3). We therefore designed our study to use tests that are sensitive to potential different mechanisms at different pregnancy stages by analyzing the quantitative trait gestational duration as well as the three dichotomous traits (early preterm birth, preterm birth, and postterm birth). We have, however, still adhered to rigorous standards for GWAS replication. Thus, while we did see two genome-wide significant association signals in the early preterm birth discovery analysis, the SNPs at these loci failed to replicate, and we do not claim that they represent genuine biological associations. One more point is that the dichotomous traits are of high clinical relevance, and results for those traits will be of interest and expected among clinically interested readers of the article. Further, we now point out in the Data Availability statement that GWAS summary statistics for all four analyzed outcomes will be made available, so that readers can interpret the results themselves.

To clarify the rationale for analyzing the quantitative trait of gestational duration as well as the three dichotomous traits, we now emphasize in the Introduction that these are clinically relevant, and include the following in the Methods section:

“The dichotomous trait analyses did not include additional individuals compared to the gestational duration analysis. However, these traits are of high clinical relevance and were therefore included in the study design. Also, including the dichotomous trait tests makes the study sensitive to potentially changing mechanisms influencing parturition at different pregnancy stages.”

2. Comparison of fetal and maternal associations in conditional models is a key part of the paper (results on Lines 167-168). But there does not appear to be a formal test to indicate if a SNP is maternal or fetal or both? Instead the authors rely on a visual inspection of effect sizes. Results for the other 6 (4) published maternal GWAS SNPs should be described in more detail in the text.

We agree that comparison of fetal and maternal associations is an important part of the article, in particular for the novel locus that we identify. Our fetal discovery stage results were based on inverse-normal transformed gestational duration and are therefore not directly comparable to the maternal results of Zhang et al 2017 (PubMed ID: 28877031), which were based on gestational duration in days. We therefore used the iPSYCH sample ($n = 51,357$) to compare fetal results in days with similar maternal

results. Before, we noted that the maternal effect estimate (0.22) was approximately half of the fetal estimate (0.37), but we did not do any formal testing. Following the reviewer's suggestion, we have included a formal test using a recently developed approach (Warrington et al. 2019 Nature Genetics (in press); preprint: <https://doi.org/10.1101/442756>). The test supports that the association is of fetal origin. The following has been added to the Results section:

"Also, we used a recently developed weighted linear model (WLM) approach³⁵ to obtain an estimate of the fetal effect adjusted for the maternal genotype and vice versa (see Methods). The WLM-adjusted fetal effect was 0.34 days (95% CI = 0.09–0.59, $P = 6.60 \times 10^{-3}$) close to the unadjusted estimate of 0.37 days, whereas the WLM-adjusted maternal effect was 0.05 days (95% CI = -0.27–0.37, $P = 0.77$)."

Furthermore, we have described results for the 4 autosomal maternal GWAS SNPs in greater detail. The relevant paragraph now reads:

"Conversely, we examined the lead variants at 4 of the 6 loci, which were reported to be significant in the maternal GWAS and were available in our meta-analysis (the remaining 2 were not autosomal)²⁸. We found evidence of association in the fetal genome at the *EBF1* ($P = 1.18 \times 10^{-6}$), *EEFSEC* ($P = 0.05$), *WNT4* ($P = 5.37 \times 10^{-5}$), and *ADCY5* ($P = 0.005$) loci. For all four loci, the direction of effects in the fetal GWAS was consistent with the published maternal GWAS results, but fetal effect size estimates were smaller (**Supplementary Table 8**)."

3. With only one confirmed GWAS hit identified, robust insights for this 2q13 locus are essential. However the functional assays for 2q13, which lies in a dense gene region, are very limited for the following reasons i) the selection of the top SNP at this locus for functional testing is based on a metric that favours wide pleiotropic actions (L175) rather than a specific function which is also plausible; ii) they overlook the 4 missense variants identified in the discovery sample because these were not significant in their much smaller follow-up exome sequencing in n=11,455, iii) very limited sample size for studies of HIC1 binding (n=4, P=0.013) and lack of a functional connection between HIC 1 and their placental expression data.

Our selection of the candidate variant by enrichment for numerous epigenetic marks in disease-specific tissues follows a well-established analytical approach in the field (Boyle et al 2012; PMID: 22955989), but we agree with the reviewer that our results do not rule out the possibility that other variants at the locus could be causal. Also, while we identify a plausible functional variant at 2q13, we agree with the reviewer that it remains to be investigated if altered HIC1 binding influences placental expression of the genes at the locus. We have clarified these points in the Discussion.

Regarding the possible effects of coding variants at the locus, we have addressed this question with greater power by including additional exome sequencing data (increasing the sample size to 18,382 individuals) and conducting joint tests of coding variants (see the response to reviewer 2's comment #1). These analyses still did not suggest that exonic variants were likely to explain the 2q13 association.

Experience tells us that understanding the biological mechanisms underlying genetic association findings can be a laborious endeavor requiring the expertise from many different research fields. A striking example is provided by the *FTO* locus for BMI and obesity. It took years of research involving a sophisticated combination of bioinformatics analyses and experiments in many different labs to identify the causal intronic *FTO* variant and understand its long-ranging effects on *IRX3* and *IRX5* expression.

We therefore think it is beyond the scope of the current study to provide a full functional understanding of the mechanisms underlying the genetic association at the 2q13 locus, and we anticipate that this will require many follow-up studies using different experimental and analytical approaches. With the robust evidence that we present for an association between fetal genetic variation at the 2q13 locus gestational duration we do, however, provide a strong foundation for such studies. We consider it a further strength of our study that we take the first steps towards understanding the molecular mechanisms underlying the genetic association.

4. The authors should report more comprehensively in the text the associations of all highlighted GWAS SNPs on continuous GA, preterm and early preterm lead SNPs.

Done. We have revised the text to include early preterm birth and preterm birth association results for rs7594852 (lead SNP for gestational duration). We now also include preterm birth and gestational duration association results for the two SNPs (rs112912841 and rs1877720) that were associated with early preterm birth in the discovery stage, but did not replicate.

5. Several of the Discovery cohorts may have excluded or undersampled preterm and early preterm infants. Comment on how the frequency of preterm (n4775) and early preterm (n1139) births in the overall sample (n84,689) compares to an unselected sample.

The overall fraction of preterm births in the discovery stage was $4,775/84,689 = 5.6\%$, but with heterogeneity among contributing cohorts, which included case/control studies of preterm birth (e.g., DNBC-PTB, GPN, MoBa_2008 with ~40–50% cases), birth cohorts (ALSPAC, NFBC, iPSYCH controls with more population representative fractions of preterm births), and cohorts where preterm births were not included (full details are given in Supplementary Table 1). Another source of heterogeneity is the varying ability among cohorts to apply the exclusion criteria (such as maternal conditions or pregnancy complications). While these sources of heterogeneity may have caused some underestimation of effect sizes at genuinely associated loci, it should not have resulted in increased false-positive rates. We have revised the paragraph on limitations in the Discussion to clarify this.

6. What was the estimated heritability of gestational age, preterm and early preterm birth based on GWAS SNPs? Only 1 significant SNP in a sample of 80,000 indicates that this trait has extremely low heritability.

We have treated the question of estimated SNP heritability for gestational duration in greater detail, using the WLM-approach to estimate heritability for fetal effects adjusted for maternal genotype and vice versa. Heritability estimates for the dichotomous outcomes (on liability scale) were not done, since these depend highly on assumptions about prevalence (which may differ a lot between contributing cohorts) and since we were also not able to disentangle fetal from maternal components for those outcomes.

We agree with the reviewer that the estimated heritability is low, also compared to estimates from family based studies. Such missing heritability has been observed for many traits and diseases, and we discuss possible underlying factors. The Methods section has been updated and new paragraphs were added to the Results and Discussion sections of the manuscript (see the response to reviewer 1's comment #1).

7. They explore the seeming non-linear effect of the 2q13 signal on gestational age on pages 103-114. The results are based on a simple visual impression. This needs formal testing for non-linear association.

We agree with the reviewer that a formal test for non-linearity of the association between gestational duration and the variant is desirable and we have followed this suggestion. A related point was raised by reviewer 1 (comment #2). Please see our response to this comment for a description of our approach to testing for differences in the strength of the genetic association over the course of gestation.

The relevant paragraphs in the Methods and Results sections have been updated, a new version of Supplementary Fig. 5 has been made, and a detailed description of the approach is given in the Supplementary Methods.

8. Line 118 - Report in the text which genes harbour the 4 missense SNPs at 2q13.

We have clarified in the main text that the two synonymous variants were in the genes *CKAP2L* and *IL1RN* and all four missense variants were in *CKAP2L*.

9. Line 161-163 - Results of fetal-maternal conditional models are key to this paper and should be reported in the text or main display rather than just citing the supplementary data.

We agree with the reviewer and we have moved the fetal-maternal results from Supplementary Table 8 to Table 2 in the main text.

Reviewers' Comments:

Reviewer #2:

Remarks to the Author:

No further comments.

Reviewer #3:

Remarks to the Author:

The authors now show convincingly that the 2q13 locus represents a specifically fetal genetic signal for gestational age. As such it is the first fetal specific contribution to gestational timing and is a noteworthy finding.

Lines 182-187: the authors now include fetal associations for the previously reported maternal GWAS for gestational age. These should be tested in conditional models to show whether these are indeed maternal signals or if there is some fetal genotype contribution.

Line 352: "the effect was strongest in later stages of pregnancy" is not accurate. In Supplementary Fig. 5, the difference in observed allele frequency increases consistently between the upper 4 quintiles of gestational age, hence the 2q13 locus appears to influence gestational age consistently across most of the spectrum of this outcome except in the lowest quintile.

Responses to Reviewer 1:

A. Variance explained: The authors have revisited these heritability estimates using appropriate approaches that consider separately the maternal and fetal genetic contributions. Unfortunately this leads to a much reduced estimate of the contribution of the fetal genome. Furthermore, they infer that the fetal genome contributes more to later variation in GA than to earlier PTB. This could be formally assessed by comparing different categories of GA to the median.

B. New results on genetic correlations Lines 201-206: Positive correlations with birth weight are to be expected and are not really a result. The other significant genetic correlations, with type of commuting and health satisfaction and others, seem to be random findings. Having showed that most of the unadjusted fetal genome contribution to GA is in fact due to its reflection of the maternal genome (Lines 199-200), the genetic correlations should ideally also be parsed into fetal and maternal contributions. This might reveal more interesting correlations.

C. Early or late biometric models: The new bootstrapping calculations appropriately address the issue of whether the 2q13 locus shifts the whole distribution of GA or acts only on certain parts of the distribution. It leaves the question as to whether then the earlier parts of the distribution are influenced more by maternal genetic factors or by other non-genetic stochastic stimuli?

NCOMMS-18-30828A

Point-by-point response to reviewers

We thank the editor and reviewers for the positive feedback and for the opportunity to improve our manuscript further.

Reviewers' comments:

Reviewer #2 (Remarks to the Author):

No further comments.

We thank the reviewer for their consideration.

Reviewer #3 (Remarks to the Author):

The authors now show convincingly that the 2q13 locus represents a specifically fetal genetic signal for gestational age. As such it is the first fetal specific contribution to gestational timing and is a noteworthy finding.

Lines 182-187: the authors now include fetal associations for the previously reported maternal GWAS for gestational age. These should be tested in conditional models to show whether these are indeed maternal signals or if there is some fetal genotype contribution.

We have revisited the mother-child pair analyses, including some additional pairs and now reaching a total of 15,588. For the top SNP rs7594852, there are slight changes to some numbers in Table 2, but the overall interpretation and conclusion remains the same. Following the reviewer's suggestion, we have also performed mother-child pair analyses for the four autosomal SNPs from the maternal GWAS of gestational duration. For all four SNPs, maternal effects adjusted for fetal genotype were stronger than fetal effects adjusted for maternal genotype, as expected. Results for those SNPs are given in the revised version of Supplementary Table 8.

Line 352: "the effect was strongest in later stages of pregnancy" is not accurate. In Supplementary Fig. 5, the difference in observed allele frequency increases consistently between the upper 4 quintiles of gestational age, hence the 2q13 locus appears to influence gestational age consistently across most of the spectrum of this outcome except in the lowest quintile.

When using the term "later stages of pregnancy", we were describing pregnancies that did not result in preterm births, but had progressed to term or beyond. Due to the skewed distribution of gestational duration in the population, this comprises most pregnancies. We have rephrased the sentence to clarify that the effect was strongest in pregnancies that went to term or beyond.

Responses to Reviewer 1:

A. Variance explained: The authors have revisited these heritability estimates using appropriate approaches that consider separately the maternal and fetal genetic contributions. Unfortunately this leads to a much reduced estimate of the contribution of the fetal genome. Furthermore, they infer that

the fetal genome contributes more to later variation in GA than to earlier PTB. This could be formally assessed by comparing different categories of GA to the median.

We appreciate that the reviewer is satisfied with our approach to separate the heritability. As expected when separating fetal from maternal contributions, the heritability estimates decrease. We agree that our results, in line with other studies referred to in the manuscript, suggest that the relative contribution of the fetal genome to variation is smaller in the earlier stages of pregnancy. However a formal assessment is problematic since the variance explained by the fetal genome is fairly low and further stratification would leave little information in the categories to be compared.

B. New results on genetic correlations Lines 201-206: Positive correlations with birth weight are to be expected and are not really a result. The other significant genetic correlations, with type of commuting and health satisfaction and others, seem to be random findings. Having showed that most of the unadjusted fetal genome contribution to GA is in fact due to its reflection of the maternal genome (Lines 199-200), the genetic correlations should ideally also be parsed into fetal and maternal contributions. This might reveal more interesting correlations.

We agree that positive genetic correlations with birth weight are to be expected and that the correlation results based on unadjusted fetal summary statistics could potentially be driven by effects in the maternal genome. Following the reviewer's suggestion, we have therefore now included genetic correlation results for WLM adjusted fetal effects and WLM adjusted maternal effects (Supplementary Table 9). The only correlation remaining significant after Bonferroni correction is between WLM adjusted maternal gestational duration and birth weight of first child.

C. Early or late biometric models: The new bootstrapping calculations appropriately address the issue of whether the 2q13 locus shifts the whole distribution of GA or acts only on certain parts of the distribution. It leaves the question as to whether then the earlier parts of the distribution are influenced more by maternal genetic factors or by other non-genetic stochastic stimuli?

This is an interesting question for future research, but we think it is beyond the scope of the current study.

Reviewers' Comments:

Reviewer #3:

Remarks to the Author:

The response to reword the conclusion on line 357-358

"The effect was strongest in pregnancies that went to term or beyond" hides the fact that there is really no association at all with preterm birth or any change in allele frequency below around 275 days = 39 weeks (in supplementary figure 5).

This difference between the significant genotype effect within the range of term and beyond GA compared to no effect on earlier GA is an important finding with relevance to the interpretation. It would be explicit to state "but not associated with risk of preterm birth" here and also in the abstract line 5.

I am happy with the responses to the other additional comments raised.

NCOMMS-18-30828C

Point-by-point response to reviewers

REVIEWERS' COMMENTS:

Reviewer #3 (Remarks to the Author):

The response to reword the conclusion on line 357-358

"The effect was strongest in pregnancies that went to term or beyond" hides the fact that there is really no association at all with preterm birth or any change in allele frequency below around 275 days = 39 weeks (in supplementary figure 5).

This difference between the significant genotype effect within the range of term and beyond GA compared to no effect on earlier GA is an important finding with relevance to the interpretation. It would be explicit to state "but not associated with risk of preterm birth" here and also in the abstract line 5.

We agree with the reviewer that it is an important result of the study that there was no association with preterm birth at the 2q13 locus, and we highlight this point in the opening paragraph of the Discussion as well as in the fifth paragraph of the Discussion.

We prefer to focus the Abstract and the concluding sentences of the article on the positive findings of the study, and have therefore not explicitly stated "not associated with risk of preterm birth" at these particular places of the manuscript. However, we have clarified the relevant sentence of the conclusion, so that it now reads:

"The effect was **observed** in pregnancies that went to term or beyond."

I am happy with the responses to the other additional comments raised.

Thank you.